# The *Drosophila* hematopoietic niche assembles through collective cell migration controlled by neighbor tissues and Slit-Robo signaling

Kara A Nelson[1,2], Kari F Lenhart[3], Lauren Anllo[1,2†], Stephen DiNardo[1,2]*

[1]Department of Cell and Developmental Biology, Perelman School of Medicine at the University of Pennsylvania, Philadelphia, United States; [2]Institute for Regenerative Medicine at the University of Pennsylvania, Philadelphia, United States; [3]Department of Biology, Drexel University, Philadelphia, United States

## eLife Assessment

This study presents **valuable** findings on the role of a well-studied signal transduction pathway, the Slit/Robo system, in the context of the assembly of the hematopoietic niche in the *Drosophila* embryo. The evidence supporting the claims of the authors is **solid**. The work will interest developmental biologists working on molecular mechanisms of tissue morphogenesis.

**Abstract** Niches are often found in specific positions in tissues relative to the stem cells they support. Consistency of niche position suggests that placement is important for niche function. However, the complexity of most niches has precluded a thorough understanding of how their proper placement is established. To address this, we investigated the formation of a genetically tractable niche, the *Drosophila* Posterior Signaling Center (PSC), the assembly of which had not been previously explored. This niche controls hematopoietic progenitors of the lymph gland (LG). PSC cells were previously shown to be specified laterally in the embryo, but ultimately reside dorsally, at the LG posterior. Here, using live-imaging, we show that PSC cells migrate as a tight collective and associate with multiple tissues during their trajectory to the LG posterior. We find that Slit emanating from two extrinsic sources, visceral mesoderm and cardioblasts, is required for the PSC to remain a collective, and for its attachment to cardioblasts during migration. Without proper Slit-Robo signaling, PSC cells disperse, form aberrant contacts, and ultimately fail to reach their stereotypical position near progenitors. Our work characterizes a novel example of niche formation and identifies an extrinsic signaling relay that controls precise niche positioning.

## Introduction

Homeostasis and repair of many organs relies on a resident stem cell population. Stem cell behavior is often coordinated by a niche (*Morrison and Spradling, 2008*)—the specific microenvironment that contains stem cells—and abnormal regulation can severely impact health by leading to tissue atrophy or tumor formation (*Chakkalakal et al., 2012*; *Ferraro et al., 2010*; *Kobielak et al., 2007*; *Walkley et al., 2007*). Thus, it is essential to understand stem cell-niche interactions. An important step in understanding the behavior of a mature niche is understanding the particular attributes it acquires—such as cellular features and the positioning of its cells with respect to each other and stem cells—and how those attributes are regulated as the niche is built during development. Yet, there are very few

*For correspondence:
sdinardo@pennmedicine.upenn.edu

Present address: †Department of Biology, East Carolina University, Greenville, United States

Competing interest: The authors declare that no competing interests exist.

studies of niche formation at the cellular level. The few well-studied niches consistently acquire tissue-specific positional and structural characteristics (*Anllo et al., 2019*; *Biggs et al., 2018*; *Gordon et al., 2020*; *Sumigray et al., 2018*). The reproducibility of these characteristics suggests that acquisition of a particular architecture is essential to niche function and emphasizes the importance of understanding how the architecture of a given niche is established.

Frequent barriers to studying niche formation include niche complexity and lack of niche cell markers (*Heitman et al., 2018*; *Pinho and Frenette, 2019*). Studying niche formation in vivo is further hindered by an inability to follow its constituent cells as they form the niche in real-time due to lack of tissue and organism transparency, and the existence of large-scale tissue movements (*Bostock et al., 2020*; *Boulais and Frenette, 2015*; *Gregg and Butcher, 2012*). The mammalian hematopoietic stem cell niche is a prime example of a difficult-to-study niche due to its location within opaque bone and its complexity—it is comprised of numerous cellular components and molecular factors (*Boulais and Frenette, 2015*; *Calvi et al., 2003*; *Pinho and Frenette, 2019*; *Zhang et al., 2003*; *Calvi et al., 2003*). Despite these limitations, some niches, such as the mammalian intestinal epithelium and hair follicle, are beginning to be studied at the cellular level (*Anllo and DiNardo, 2022*; *Díaz-Torres et al., 2021*; *Gupta et al., 2019*; *Pentinmikko et al., 2022*; *Sumigray et al., 2018*). However, investigations about establishment of these well-characterized niches still suffer from challenges including limited real-time imaging and lack of tissue-specific manipulations.

To overcome these limitations and gain insight into the mechanisms that drive niche formation, we have investigated the development of the niche that supports the *Drosophila* larval hematopoietic organ: the lymph gland (LG). The cells that constitute this niche, called the posterior signaling center (PSC), and the markers that label it are known (*Crozatier et al., 2004*; *Lebestky et al., 2003*; *Mandal et al., 2007*). Additionally, the PSC resides under a thin epidermal covering, making it amenable to live-imaging with high spatial resolution, as we show here. The dynamic information afforded by live-imaging can reveal the mechanism of migration and implicate nearby tissues as a source for guiding signals. These advantages, paired with facile *Drosophila* genetics, make for a powerful experimental system in which PSC formation can be visualized in vivo and the underlying mechanisms probed.

Minimal information exists about the formation of the PSC, as most analyses of the lymph gland have focused on the steady-state operations of late larval stages when the gland is mature and most accessible due to its larger size. While the mature LGs are comprised of multiple pairs of lobes (*Koranteng et al., 2022*; *Shrestha and Gateff, 1982*), we use 'lymph gland' to refer to only one of the pair of thoroughly characterized, bilaterally symmetric 'primary' lobes. In late larval stages, the LG contains thousands of cells, organized into multiple zones: the medullary, intermediate, and cortical zones, and the PSC (*Banerjee et al., 2019*; *Jung et al., 2005*; *Lanot et al., 2001*). The PSC regulates the adjacent hematopoietic progenitors of the medullary zone; these progenitors are progressively differentiated from the innermost region of the LG, radially outward (*Cho et al., 2020*; *Jung et al., 2005*; *Luo et al., 2020*; *Mandal et al., 2007*). The mature, terminally differentiated hemocytes reside at the outermost cortex of the gland, in the cortical zone, and between these zones are the cells of the intermediate zone (*Jung et al., 2005*; *Krzemien et al., 2010*; *Spratford et al., 2021*). Recent scRNA-seq analyses have identified new types of hemocytes in the LG (*Cho et al., 2020*; *Girard et al., 2021*), but the three main types are analogous to vertebrate myeloid cells and include crystal cells, plasmatocytes, and lamellocytes, which are responsible for wound healing and innate immunity (*Banerjee et al., 2019*; *Lanot et al., 2001*; *Rizki and Rizki, 1992*).

The larval PSC facilitates wound repair, immune response, and homeostasis and does so by performing two key functions: maintaining progenitors and inducing progenitor differentiation (*Baldeosingh et al., 2018*; *Crozatier et al., 2004*; *Krzemień et al., 2007*; *Mandal et al., 2007*; *Ramesh et al., 2021*). PSC positioning is such that it contacts both the least differentiated progenitors and mature hemocytes (*Baldeosingh et al., 2018*; *Jung et al., 2005*), a seemingly prime location where the PSC is poised to both implement regulation and receive feedback, thereby maintaining homeostasis in the LG. For example, upon immune challenge, the PSC senses and responds to the threat by instructing differentiation of progenitors (*Khadilkar et al., 2017*; *Louradour et al., 2017*; *Sinenko et al., 2012*). The PSC engages multiple signaling pathways to execute its roles and relies on cell biological characteristics of its component cells, such as occluding or gap junctions, to do so (*Baldeosingh et al., 2018*; *Ho et al., 2023*; *Khadilkar et al., 2017*; *Mandal et al., 2007*; *Sinenko et al., 2009*).

PSC functionality is integral for organism health, as PSC loss can cause precocious differentiation of progenitors under homeostatic conditions and inability to produce lamellocytes under immune challenge conditions (*Baldeosingh et al., 2018*; *Krzemień et al., 2007*; *Mandal et al., 2007*). PSC-regulated homeostatic maintenance of the larval LG is crucial, as the LG will ultimately rupture, releasing its constituents into circulation to contribute to the hematopoietic pool in pupal and adult stages (*Grigorian et al., 2011*; *Holz et al., 2003*). Furthermore, recent publications suggest that the PSC itself adopts new functions upon LG rupture – PSC cells become highly motile and phagocytic and are capable of transdifferentiating into lamellocytes or plasmatocytes upon immune challenge (*Boulet et al., 2021*; *Hirschhäuser et al., 2023*).

Though the late larval PSC is comprised of about 30–50 cells (*Ho et al., 2021*; *Morin-Poulard et al., 2016*; *Tokusumi et al., 2015*), it is initially specified as only about five cells (*Mandal et al., 2007*). The expansion of PSC cell number takes place only after the five cells become organized into the PSC at the posterior of the developing LG during late stages of embryogenesis (*Mandal et al., 2007*). How the PSC becomes consistently positioned after specification is unknown and is our focus here. The LG itself is specified about mid-way through embryogenesis, around stage 12, from cardio-genic mesoderm as three cell clusters located laterally—one in each thoracic segment (schematized in *Figure 1A*, left; *Crozatier et al., 2004*; *Mandal et al., 2004*; *Mandal et al., 2007*). The PSC is specified from the posterior-most LG cells by expression of Antennapedia and the silencing of Homothorax (*Mandal et al., 2007*). A transcription factor, Collier, the *Drosophila* ortholog of mammalian early B-cell factor, is initially expressed in the entire LG primordium but becomes restricted to the PSC and is necessary for its maintenance and function (*Crozatier et al., 2004*; *Krzemień et al., 2007*). From fixed preparations, its known that the three LG clusters coalesce along the anterior-posterior axis before reaching the dorsal midline. At the end of embryogenesis, the cluster of PSC niche cells resides at the posterior of each bilaterally symmetric lymph gland; each gland flanks the dorsal vessel—the *Drosophila* heart—at the dorsal midline (schematized in *Figure 1A*, right; *Crozatier et al., 2004*; *Mandal et al., 2007*). Apart from this description nothing is known about how the PSC is built, and no live-imaging of the PSC has been conducted during embryogenesis. Despite the probable importance of PSC positioning to later function, the mechanism of migration, the signals guiding migration, and interactions among constituent cells—each integral aspects of organogenesis—are entirely unknown.

We have conducted the first in vivo live-imaging of PSC formation, visualizing migration of PSC cells from their origin, lateral in the embryo, to their final, dorsal position at the posterior of the LG. We find that a prominent, regulated feature of this niche is the migration of its constituent cells as a collective. Live-imaging also revealed neighboring tissues that could feasibly signal to influence PSC migration as well as its positioning in the LG. Using genetic ablation and mutant analyses, we have identified that both the dorsal vessel and visceral mesoderm are required for proper positioning of the PSC. In addition, mutant analyses and tissue-specific challenges revealed an intricate web of Slit-Robo signaling in and between these tissues that is essential to establish positioning of PSC cells as a coalesced, compact collective at the LG posterior. Finally, we use live-imaging to show that Slit signaling is essential to maintain collectivity of PSC cells and properly position these cells during their migration. Taken together, we have uncovered a new example of collective cell migration, revealed some of its mechanistic underpinnings, and implicated a signaling relay required to properly build and position this niche.

## Results

### Live-imaging reveals dynamics of PSC formation

While PSC formation is undoubtedly a dynamic process, current knowledge is derived exclusively from static timepoints using fixed embryos (*Figure 1A*). Thus, we sought to visualize the entirety of PSC migration in real time. To identify and track PSC cells live, we first confirmed their location at various time points in fixed embryos using the accepted markers, Antp and Odd, where co-expression demarcates PSC cells. We characterized PSC positioning in stages ranging from migration onset, stage 14, until migration completion, stage 16 (schematized in *Figure 1B'–F'*). Prior to migration, the PSC appeared as a compact cluster of cells (*Figure 1B*, green underline), it appeared more elongated through mid-migration (*Figure 1C–E*), and ultimately it assumed a compact, clustered organization at the posterior of the lymph gland (*Figure 1F*). During migration and upon its completion, the PSC

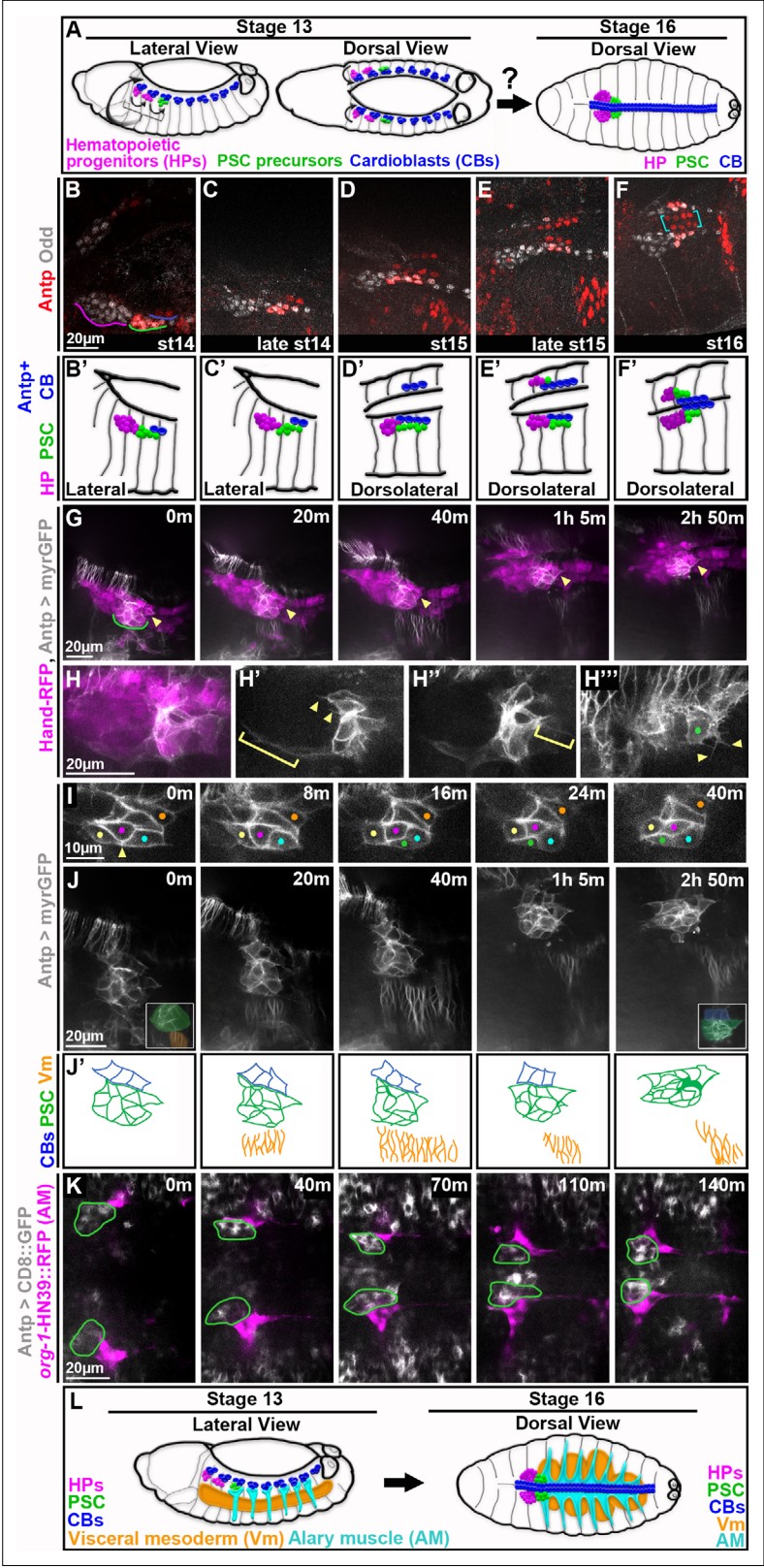

**Figure 1.** Live-imaging reveals dynamics of PSC migration and presence of nearby muscles. All images are oriented with anterior to the left and posterior to the right. (**A**) Schematic depicts prior knowledge of PSC migration. Prospective cells of lymph glands (hematopoietic progenitors, magenta, and PSC, green) and cardioblasts (blue) of the dorsal vessel are specified as distinct clusters prior to migration onset around stage 13

*Figure 1 continued*

(left). Around stage 16 (right) lymph gland clusters have coalesced and flank the dorsal vessel at the dorsal midline; bilaterally symmetric counterparts are aligned. (B-F) Fixed $w^{1118}$ embryos stained for Antp and Odd to label the PSC. (**B**) Stage 14 PSC (green) is lateral in embryo, tightly clustered, and flanks 2 Antp + CBs (blue). (**C**) Late stage 14 PSC is more dorsal, flanking 2 Antp + CBs. (**D**) The stage 15 PSC is even more dorsal, elongated, and flanks 4 Antp + CBs; contralateral Antp + CBs in view. (**E**) Late stage 15 PSC is more medial and compact. (**F**) Stage 16 LGs and CBs at final position at dorsal midline, right LG partially in view. PSC is compact and coalesced at LG posterior, flanking Antp + CBs. Antp + CBs (yellow brackets) are neatly aligned in stereotypical 2x4 organization. (**B'-F'**) Schematics depicting relevant cell types in **B-F**: hematopoietic progenitors (magenta), PSC (green), and Antp + cardioblasts (blue). (**G-J**) Live-imaging stills from Hand-RFP,Antp-GAL4,10xUAS-myr:GFP embryos; RFP in magenta and GFP in white. (**G**) Timelapse shows dorsolateral view of PSC (magenta and white cells above green underline and below string of CBs, arrows) migrating as a collective from migration onset (0 m), when the PSC is lateral, to migration completion (around 2h50m), when the PSC is dorsal. Arrowhead indicates fixed positioning of the posterior-most PSC cell in the collective. (**H-H'''**) Live-imaging stills show PSC protrusions during mid-migration. (**H**) Merge shows positional relation of PSC (white and magenta) and its protrusions to rest of lymph gland and CBs (along top of image). (**H'-H'''**) Only GFP channel. Progressively shallower z-slices of same PSC. (**H'**) Anterior protrusions; some short (arrowheads) and one long (bracket) that encases lymph gland. (**H''**) Posterior protrusion (bracket). (**H'''**) Green dot indicates dorsal-most PSC cell with short, branching protrusions (arrowheads). Much of image contains Antp +epidermis. (**I**) Live-imaging stills of GFP channel only show PSC cells shifting position within collective. Initially connected yellow and cyan PSC cells (0 m, arrowhead) become separated by intercalation of magenta and green PSC cells while the posterior-most, dorsal vessel-adjacent PSC cell (orange dot) remains at a fixed position in the collective. (**J**) Same embryo as (**G**), GFP channel only. For the entire timelapse PSC cells migrate dorsally in close association with concurrently migrating neighbor tissues, CBs and Vm. Posterior-most PSC cell (arrowhead) maintains its dorsal vessel-adjacent position throughout. 0 min inset shows PSC proximity to Vm and 2hr 50min inset shows PSC proximity to CBs. (**J'**) Schematics depict relevant cell types for each panel of J time series: cardioblasts (blue), PSC (green), and visceral mesoderm (orange). (**K**) Live-imaging series from dorsal vantage point shows A1 AMs (magenta; labeled with *org-1*-HN39::RFP) contact PSCs (green outlines; labeled by Antp >CD8::GFP) throughout the timelapse as both tissues migrate to dorsal midline. (**L**) Schematics depict the relative positioning of the tissues identified to be near the PSC at migration onset (left) and completion (right).

The online version of this article includes the following video and figure supplement(s) for figure 1:

**Figure supplement 1.** ECM components surround the PSC and AM encases PSC cells.

**Figure supplement 2.** LG and CBs migrate independently of dorsal closure.

**Figure 1—video 1.** Timelapse imaging of wildtype PSC migration from dorsolateral vantage visualized with Hand-RFP, Antp>myrGFP.

https://elifesciences.org/articles/100455/figures#fig1video1

**Figure 1—video 2.** Live-imaging of PSC collective visualized with Antp>myrGFP from dorsolateral vantage point during migration.

https://elifesciences.org/articles/100455/figures#fig1video2

**Figure 1—video 3.** Timelapse imaging of wildtype PSC migration from dorsolateral vantage visualized with Hand-RFP,686 Antp>myrGFP (left) and Antp>myrGFP only (right).

https://elifesciences.org/articles/100455/figures#fig1video3

**Figure 1—video 4.** Magnified version of *Figure 1—video 3*.

https://elifesciences.org/articles/100455/figures#fig1video4

Stills from timelapse shown in *Figure 1J*.

**Figure 1—video 5.** Live-imaging with dorsal view of PSC and AM migration visualized with Antp>mCD8:GFP and org-1-HN39::RFP, respectively, beginning at migration onset.

https://elifesciences.org/articles/100455/figures#fig1video5

---

flanked Antp + cardioblasts of the dorsal vessel (*Figure 1F*, brackets). Though not explored further here, ECM components, Perlecan and Viking, were expressed around the PSC collective (*Figure 1—figure supplement 1A, B*), including between the PSC and the adjacent cardioblasts (CBs) of the dorsal vessel.

The prospective cells of the LG and CBs lie just under a sheet of epidermis that itself undergoes dramatic movement in late-stage embryos, concurrent with LG and CB migration. Leading edge (LE) epidermal cells move to the dorsal midline during dorsal closure, which is complete by stage 16. It has previously been shown that CBs migrate independently of LE epidermal cells (*Haack*

*et al., 2014*; *Balaghi et al., 2023*). To address whether the LG/PSC also migrates independently, we assessed LG positioning in stage 17 mutants in which dorsal closure had stalled. Both heterozygous and homozygous *hlh54f*$^{598}$ mutants (*Figure 1—figure supplement 2A, B, C*, respectively) exhibited defects in closure, evident by epidermal leading edge (LE) cells that stalled at lateral positions (LE indicated by cyan lines). Embryos stained with p-Tyr to label cell membranes and Odd to label the LG were scored for position of the LGs relative to the stalled LE. In over two-thirds of cases the LG had migrated past the stalled LE (*Figure 1—figure supplement 2D*), indicating that LG migration is generally uncoupled from this large-scale epidermal movement, similar to the case for CBs.

To our knowledge, a sufficiently bright, stable marker that labels only PSC cells during migration stages does not exist. Thus, for live-imaging we combined Hand-RFP, which marks lymph gland (including the PSC) and CBs, with a myristoylated GFP driven by Antp-GAL4 (Antp >myrGFP) to mark the PSC but not other lymph gland cells (*Figure 1G–J*). We performed timelapse imaging on embryos beginning after PSC specification (stage 13) until the PSC completed migration and morphogenesis (stage 17). We observed that the PSC moved as a collective for the entire migration (*Figure 1G*; *Figure 1—video 1*). During migration many PSC cells extended protrusions—some reached far anteriorly, encasing the lymph gland (*Figure 1H'*, bracket) while others were shorter or branched (*Figure 1H''''*, arrowheads). The protrusions exhibited no apparent directional bias but were more common at non-dorsal vessel-adjacent surfaces of the collective. During migration some cells shifted positioning with respect to one another (*Figure 1—video 2*). For example, two lateral PSC cells that abut early on (*Figure 1I*, 0m, arrowhead) become separated by interdigitating cells (*Figure 1I*, 8-40m, magenta and green dots). In general, however, most cells remained at relatively fixed positions within the collective—particularly the posterior-most, dorsal vessel-adjacent PSC cells (*Figure 1I*, orange dots and *Figure 1G*, arrowheads).

Overall, the features observed during live-imaging lead us to characterize PSC formation as collective cell migration. The intricate, dynamic extensions we observed on PSC cells reflect an active actin-based cytoskeleton—a common theme amongst migrating cell collectives, where these extensions are often used for sensing environmental guidance cues. Most importantly, even as some PSC cells shifted their position relative to one another, the group accomplished directional movement while maintaining coalescence. The persistence of PSC-to-PSC cell contacts despite internal movements suggests coordination within the group to orchestrate remodeling of cell adhesion such that a unified directional movement is achieved. Altogether, these features are consistent with this as an example of collective cell migration, and suggest the importance of collectivity in building the PSC.

## Visceral, cardiac, and alary muscles are near the PSC throughout its migration

Crucially, live-imaging also uncovered that the PSC was within signaling distance of various muscles for the entirety of its migration. It is well-documented that muscles can supply positional information to nearby tissues (*Anllo and DiNardo, 2022*; *Scimone et al., 2017*; *Witchley et al., 2013*), and our imaging indicated that visceral muscle (or, visceral mesoderm; Vm) and CBs were each near the PSC. We live-imaged with a dorsolateral vantage point which revealed that the PSC migrated dorsally in synchrony with CBs and Vm (*Figure 1J*; *Figure 1—video 4*). The PSC remained laterally affixed to Antp +CBs (*Figure 1J'*, blue) and slightly dorsal to, but not contacting, Antp +Vm (*Figure 1J'*, orange) throughout migration.

Based on segmental positioning, we suspected the PSC was also near alary muscle (AM)—segmentally repeating muscles that attach internal organs to the body wall. We live-imaged PSC and AM migration from a dorsal vantage point with Antp >CD8:GFP and *org-1*-HN39::RFP, a marker with expression restricted to AMs (*Figure 1—video 5*). Indeed, the first AM was immediately lateral to the PSC for the entirety of their synchronized migration to the dorsal midline (*Figure 1K*, PSCs outlined in green). We confirmed this proximity by fixing *org-1*-HN39::RFP embryos and staining for Antp, which revealed that at late stages, the A1 AM ensheathed the PSC (*Figure 1—figure supplement 1D*), with some fibers encasing individual PSC cells (*Figure 1—figure supplement 1E*).

Thus, live-imaging provided invaluable insights, advancing our knowledge of PSC formation from that attainable through analysis of fixed preparations (*Figure 1A*), by revealing the mode of PSC migration and the spatiotemporal dynamics of the PSC relative to nearby tissues (*Figure 1L*). The

sustained proximity of alary muscles, visceral mesoderm, and cardioblasts to the PSC made each a strong candidate for influencing PSC positioning.

## Vm and CBs are required for proper PSC positioning

Having identified multiple candidates, we next investigated whether any or all of these muscles were necessary for proper PSC positioning. To test a role for AM, we ablated it by expressing the pro-apoptotic factor, *grim*, using the AM-specific AME$_r$-GAL4. Ablation was successful, evidenced by lack of GFP-labeled AMs (*Figure 2B*) compared to controls (*Figure 2A*). Based on our characterization of the wildtype PSC (*Figure 1*), we developed criteria to score PSCs at the culmination of their migration as positioned 'normally' or 'abnormally'. A 'normal' PSC must be (1) coalesced within one nuclear diameter of one another, (2) adjacent to the dorsal vessel, and (3) at the same dorsal-ventral position as the posterior-most progenitors of the lymph gland. Surprisingly, AM ablation had no impact on PSC positioning (*Figure 2C*) nor the total number of PSC cells (*Figure 2—figure supplement 1*), indicating that AMs are not required for proper PSC formation.

To determine whether Vm was involved in forming the PSC, we examined *biniou* mutants in which the Vm is genetically ablated (*Zaffran et al., 2001*). In controls, the Vm is apparent at stage 12 as columnar, Fas3+ cells (*Figure 2D*, brackets). By stage 16, Vm has surrounded the gut and constricted it into sections (*Figure 2E*, arrow is first Vm constriction that separates the first two gut sections). By contrast, in *bin* mutants, minimal Fas3+ Vm was detected at stage 12 (*Figure 2G*), and it was absent at stage 16 (*Figure 2H*). The majority of *bin$^{R22}$* mutants and *bin$^{R22}$ /bin$^{S4}$* transheterozygous mutants had abnormal PSC positioning (*Figure 2I*, arrowheads) compared to heterozygote controls with normal PSC positioning (*Figure 2F*, outlines; quantified in 2 J). We detected no Bin protein in PSC cells (*Figure 2—figure supplement 1B*), indicating PSC positioning defects in *bin* mutants did not originate PSC-intrinsically but rather were caused by lack of Vm.

To confirm a role for Vm in PSC formation, we ablated Vm by expressing the proapoptotic gene, *hid*, using a Vm-specific driver, bap-GAL4. Stage 11 sibling controls displayed characteristic organization of Vm precursors (*Figure 2K*): segmentally repeating mounds of fusion competent myoblasts (FCMs; between green and magenta lines) atop columnar founder cells (FCs; below magenta line). Stage 11 ablated embryos were largely missing FCMs (*Figure 2N*, brackets). Although most FCs were present at this stage, they exhibited substantially reduced Bin expression compared to controls (*Figure 2— figure supplement 1C and D*, quantified in 1E), suggesting improper differentiation. By stage 16 no Vm was detectable (*Figure 2O*). Vm-ablated embryos had abnormal PSC positioning (*Figure 2P*, arrowheads) more often than sibling controls (*Figure 2M*, quantified in 2Q). Taken together, these analyses definitively establish that an extrinsic cue(s) provided by Vm governs PSC positioning.

To address when the cue might be delivered, we analyzed *jelly belly* mutants and *bagpipe* hypo-morphs, *bap$^{208}$*. In both backgrounds, Vm precursors are present initially at stage 11 but do not differentiate or migrate properly from stages 13–16 (*Figure 2—figure supplement 1* and *Englund et al., 2003*; *Lee et al., 2003*; *Weiss et al., 2001*). In both mutants, PSC positioning was unaffected (*Figure 2—figure supplement 1K, N, and P*; quantified in L and Q), revealing that early signaling from Vm is required for PSC positioning.

Finally, to assess whether cardioblasts of the dorsal vessel impact PSC positioning, we expressed *hid* or *grim* with the CB-specific driver, tinCΔ4-GAL4. We used conditions that restricted defects to CBs and left Vm unaffected (see Materials and methods). Whereas controls had uninterrupted strings of Mef2-labeled CBs (*Figure 2R*), the manipulated embryos had significant dorsal vessel ablation, evident by gaps in CBs (*Figure 2S*, bracket). We elected to analyze stage 14 and 15 embryos because these presented with appreciable ablation over larger distances compared to stage 16 onwards. Oftentimes both sides of the bilaterally symmetric dorsal vessel were ablated; sometimes only one side was ablated. For a given instance of dorsal vessel ablation, we documented whether the ipsilateral PSC was positioned normally. While the PSC occasionally appeared mis-positioned in non-ablated controls, for ablated dorsal vessel the ipsilateral PSC was mis-positioned more frequently (*Figure 2T*). These data suggest that dorsal vessel CBs influence PSC formation.

## Slit-Robo signaling is required for proper PSC formation

To identify a cue necessary for positioning the PSC, we sought a signal that is expressed in early Vm and/or CBs. The secreted glycoprotein Slit is known to be expressed in both tissues and has been

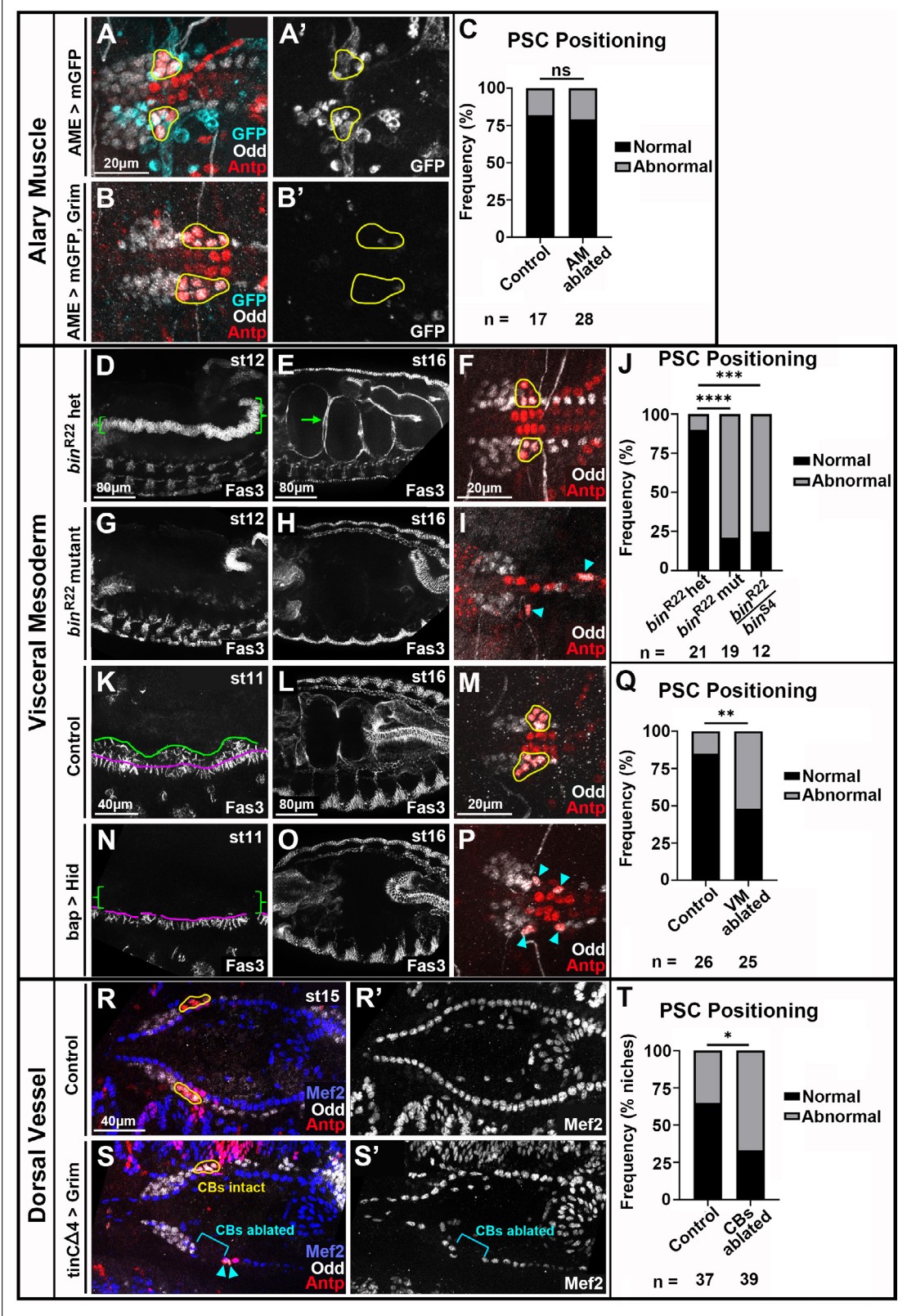

**Figure 2.** Visceral mesoderm and cardioblasts are required for PSC formation. All PSCs, co-labeled by Antp and Odd, are viewed dorsally from st16 or 17 embryos unless otherwise noted. (**A, B**) LGs with normally-positioned PSCs outlined in yellow from control (**A**) and AM-ablated (**B**) embryos with AMs labeled by AME-Gal4 driven mCD8:GFP. (**A', B'**) GFP channel only with PSC outlines overlaid. (**C**) PSC positioning quantification. (**D, E**) bin heterozygotes labeled with Fas3, lateral views. (**D**) Green bracket indicates st12 Vm. (**E**) st16 with green arrow indicating first Vm constriction that

*Figure 2 continued on next page*

*Figure 2 continued*

segregates two sections of gut. (**F**) bin heterozygote LGs with normal PSC positioning. (**G, H**) lateral views of bin mutants with minimal (**G**, st12) and absent (**H**, st16) Vm. (**I**) bin mutant LGs with dispersed PSCs (cyan arrowheads). (**J**) PSC positioning quantification including analysis of binR22/binS4 transheterozygous mutants. (**K-P**) control embryos (**K-M**) compared to Vm-ablated embryos (**N-P**). (**K**) Lateral view of normal st11 Vm labeled by Fas3. Founder cells are below magenta line and fusion competent myoblasts are between magenta and green lines. (**L**) Dorsal view of normal st16 Vm. (**M**) Normal PSC positioning, yellow outlines, in control. (**N**) Fusion competent myoblasts absent, green brackets, in st11 Vm ablated embryo. Most founder cells present with occasional gaps. (**O**) Vm absent in st16 Vm ablated embryo. (**P**) Abnormal PSC positioning, cyan arrowheads. (**Q**) PSC positioning quantification. (**R, S**) st15 LGs with cardioblasts labeled by Mef2 in control (**R**) and CB-ablated embryos (**S**). (**R**) Normal PSC positioning, yellow outlines. (**S**) CBs ablated on left side, bracket, and corresponding PSC is mis-positioned, arrowheads, while right side CBs are intact and the R PSC is positioned normally, yellow outline. (**R', S'**) Mef2 channel only. (**T**) Quantification comparing PSC positioning in control embryos to positioning of PSCs with ablated ipsilateral CBs. Scale bars as indicated. Ns = not significant, *p<0.05, **p<0.01, ***p<0.001, ****p<0.0001, Fisher's Exact test. Sample sizes as indicated.

The online version of this article includes the following source data and figure supplement(s) for figure 2:

**Figure supplement 1.** PSC analysis under various manipulations.

**Figure supplement 1—source data 1.** PSC cell count in Control and AM-ablated cases.

**Figure supplement 1—source data 2.** Bin fluorescence intensity measurements in Control and Ablated.

---

implicated as a positional cue in multiple developmental contexts (*Anllo and DiNardo, 2022*; *Kidd et al., 1999*; *Kolesnikov and Beckendorf, 2005*; *MacMullin and Jacobs, 2006*; *Rothberg et al., 1990*). We confirmed that Slit is detectable in both Vm and CBs at relatively early stages of PSC migration (*Figure 3A and B*). Importantly, *sli* mutants exhibited mis-positioned PSCs more frequently compared to controls (*Figure 3C and D*, quantified in 3E), indicating a requirement for Slit in properly positioning the PSC.

To confirm that PSC cells can respond to Slit ligand, we examined the expression of the three Robo class receptors. We found evidence for only Robo1 and Robo2 expression in the PSC (*Figure 3—figure supplement 1A, B*). Thus, to test the requirement for Robo signaling, we analyzed single and double *robo1* and *robo2* mutants. For both single mutants, about 50% of embryos had abnormal PSC positioning (*Figure 3I and J*, arrowheads). About 75% of double mutants had abnormal PSCs (*Figure 3K*), a frequency which matches that of *sli* mutants. These results demonstrate that Slit signaling through Robo1 and Robo2 is required for properly positioning the PSC.

## Dorsal vessel-derived Slit signaling is required for PSC positioning

To test whether CBs were a source of Slit important for PSC positioning, we used tinCΔ4-GAL4 to prevent either production or release of Slit specifically from dorsal vessel using RNAi or sequestration via overexpression of the Robo1 receptor, respectively. We confirmed that two independent Slit RNAi's eliminated nearly all Slit accumulation at CBs (*Figure 3O', and Q'*), while Robo overexpression resulted in more pronounced Slit accumulation on CB membranes (*Figure 3S*) compared to control CBs (*Figure 3M'*). PSCs were abnormally positioned more frequently under all challenge conditions (*Figure 3N, P and R*) compared to controls (*Figure 3S*). Thus, Slit produced from CBs is essential to properly position the PSC.

## Vm-derived Slit signaling is required for PSC positioning

The additional requirement for Vm in PSC positioning and the presence of Slit expression in Vm suggested that Vm-derived Slit might also regulate PSC formation. To block Slit produced by Vm from signaling to PSCs, we sequestered Slit specifically in Vm using bap-GAL4 to overexpress Robo1. This manipulation prevents diffusion of Slit away from the tissue, as evidenced by distinct accumulation of Slit on Vm membranes (*Figure 3Z*) compared to diffuse Slit puncta in control Vm (*Figure 3W*). Under this condition, we indeed found that PSCs were mis-positioned significantly more often (*Figure 3AA*; quantified in 3BB). Taken together, these data suggest that Vm and CB-derived Slit signaling are both required for PSC positioning.

## Vm influences dorsal vessel positioning

While the data thus far supported a simple model wherein Slit secreted from Vm and CBs acts directly on Robo receptors expressed by PSC cells to influence PSC positioning, further investigation revealed a more complex situation. Autocrine Slit-Robo signaling among dorsal vessel cells is important for proper polarity and organization of the vessel (*Medioni et al., 2008*; *Qian et al., 2005*;

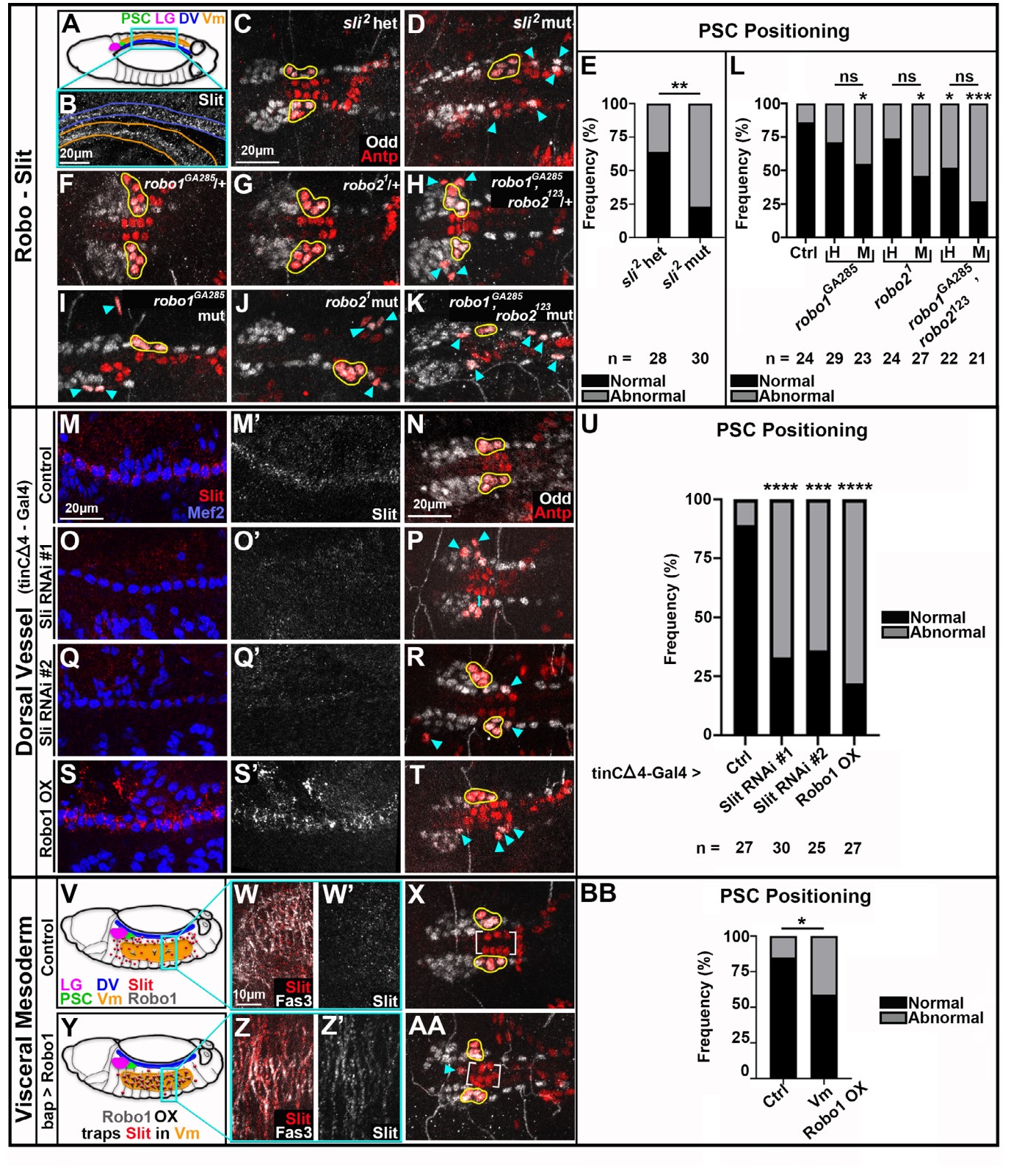

**Figure 3.** Slit from Vm and CBs signals through Robo for PSC positioning. All PSCs, co-labeled by Antp and Odd, are viewed dorsally from st16 or 17 embryos. Normally coalesced and positioned PSC cells are outlined in yellow; dispersed PSC cells indicated by cyan arrowheads (**A**) Schematic of dorsal st14 embryo. Relevant tissues are indicated; Vm is ventral to the rest. (**B**) Slit expression in st14 CBs, blue outline, and Vm, orange outline; these tissues are in reversed orientation from the schematic due to embryo tilt, and the projection necessary to have both tissues in view. (**C-E**) Analysis of

*Figure 3 continued on next page*

*Figure 3 continued*

PSC positioning in *sli²* heterozygotes (**C**, normal PSCs) and *sli²* mutants (**D**, dispersed PSCs), with frequency of positioning phenotype quantified in (**E. F-L**) Analysis of PSC positioning under various *robo* depletion scenarios. *robo1* heterozygote (**F**) and *robo2* heterozygote (**G**) with normal PSCs. (**H**) *robo1,robo2* double heterozygote with abnormal PSCs. *robo1* single (**I**), *robo2* single (**J**), and *robo1, robo2* double (**K**) mutants with abnormal PSCs; frequency of positioning phenotype quantified in L (**'H'** is heterozygote, ('**M**' is mutant). (**M-U**) Analysis of PSC positioning when Slit signaling from CBs is compromised; frequency of PSC positioning phenotype quantified in (**U. M, O, Q, S**) Slit expression in Mef2 labeled cardioblasts of controls (**M**) and tinCΔ4-GAL4 driven Slit RNAi knockdown (**O, Q**) or Robo1 overexpression (**S**) embryos. (**M', O', Q', S'**) Slit channel only. (**N**) Control with normal PSCs. (**P, R, T**) Dispersed PSCs upon compromised Slit signaling from CBs. V-BB Analysis of PSC positioning when Slit signaling from Vm is compromised; frequency of PSC positioning phenotype quantified in BB. (**V, Y**) Schematics of dorsolateral st14 control (**V**) or bap-GAL4 driven Robo1 overexpression (**Y**) embryos with relevant tissues and proteins indicated. (**W**) Diffuse Slit expression in Fas3-labeled Vm of st14 control embryo. (**Z**) Slit trapping at Fas3-labeled Vm membranes in bap-GAL4-driven Robo1 overexpression embryo. (**W', Z'**) Slit channel only. (**X**) Normally positioned PSCs and Antp+ CBs (brackets) in control. (**AA**) Abnormally positioned PSC and Antp+ CBs (brackets) in bap-GAL4-driven Robo1 overexpression embryo. Scale bars as indicated. Ns = not significant, *p<0.05, **p<0.01, ***p<0.001, ****p<0.0001, Fisher's Exact test. Sample sizes as indicated.

The online version of this article includes the following figure supplement(s) for figure 3:

**Figure supplement 1.** Robo1 and Robo2 are expressed by PSC cells.

*Santiago-Martínez et al., 2008*). Intriguingly, in experiments where Slit was sequestered in Vm, we noticed defects in the organization of the cardioblasts (*Figure 3AA*, brackets; compare to control, 3 X brackets). This suggested that signaling from Vm was required for organizing the dorsal vessel—a possibility that had not been previously explored. To test this, we examined the dorsal vessels of *bin* mutants which lack Vm (*Figure 2H*). We identified multiple phenotypes, two of which we termed 'twisted' and 'sunken'; the dorsal vessel in *Figure 4B* exhibits both phenotypes. We scored a dorsal vessel as 'twisted' (*Figure 4B*, bracket) when there was an apparent kink in the vessel such that the left or right side sat directly atop the other; thus, when viewing a single z-slice, a span of contralateral CBs appear to be missing since they are displaced above or below that focal plane. We defined a sunken dorsal vessel (*Figure 4B' and B"*) as those cases where the whole dorsal vessel, or part of it, was displaced ventrally, deeper into the embryo. Such a case is evident in the *Figure 4B* mutant dorsal vessel, which spans 8 μm of depth, compared to the sibling control dorsal vessel, which spans only 4 μm. Both phenotypes were significantly more frequent in *bin* mutants compared to sibling controls (*Figure 4C*). Together, the dorsal vessel defects observed upon removal of the Vm or from trapping Slit on Vm cells reveal a previously unrecognized role for Vm in dorsal vessel formation.

## PSC positioning requires Robo signaling in CBs and in PSCs

Our results indicated that lack of Vm causes abnormal dorsal vessel positioning, and that the dorsal vessel is required for PSC positioning. In addition, our live-imaging revealed that some PSC cells are affixed to CBs of the dorsal vessel (*Figure 1*). Thus, it became important to test whether all PSC phenotypes might be 'passive', explained by PSC attachment to a malforming dorsal vessel. Alternatively, the PSC defects could reflect a requirement for Robo activation directly in PSC cells. Without a PSC-specific driver to directly test the latter, we re-examined *robo1,robo2* double mutants, this time including an additional marker that allowed us to observe PSC and CB defects independently and score for correlation between the two. If defective PSC positioning mostly correlated with defective CB positioning this would suggest a passive effect on PSCs by CBs of the dorsal vessel (*Figure 4D*, middle schematic). By contrast, if PSC cells were mis-positioned without a similarly mis-positioned CB nearby, this would suggest a requirement for intrinsic activation of Robo in PSC cells (*Figure 4D*, rightmost schematic).

The CB marker, *svp*-lacZ, labels two CBs in each hemisegment (*Figure 4E*, two hemisegments shown; *Lo and Frasch, 2001*). Normally, the PSC is compact and located adjacent to the compact Antp +CBs (*Figure 4E*, PSCs outlined). Svp co-labels the last of the four strongly Antp +CBs, and it labels the immediately posterior Antp- CB (*Figure 4E*, brackets). At the end of migration, Svp-LacZ +CBs, like all CBs, are well-aligned with their bilaterally symmetric counterparts. Thus, including Svp-LacZ in our analysis served as a useful registration marker that afforded the ability to detect bilateral matching, as well as shifting of CBs or the PSC along the anterior-posterior axis (*Figure 4E–G*, arrows indicate second set of Svp +CBs).

In *robo1,robo2* double mutants, we found that mis-positioning of CBs and the ipsilateral PSC occurred more frequently compared to double heterozygotes (*Figure 4H*). We then further analyzed

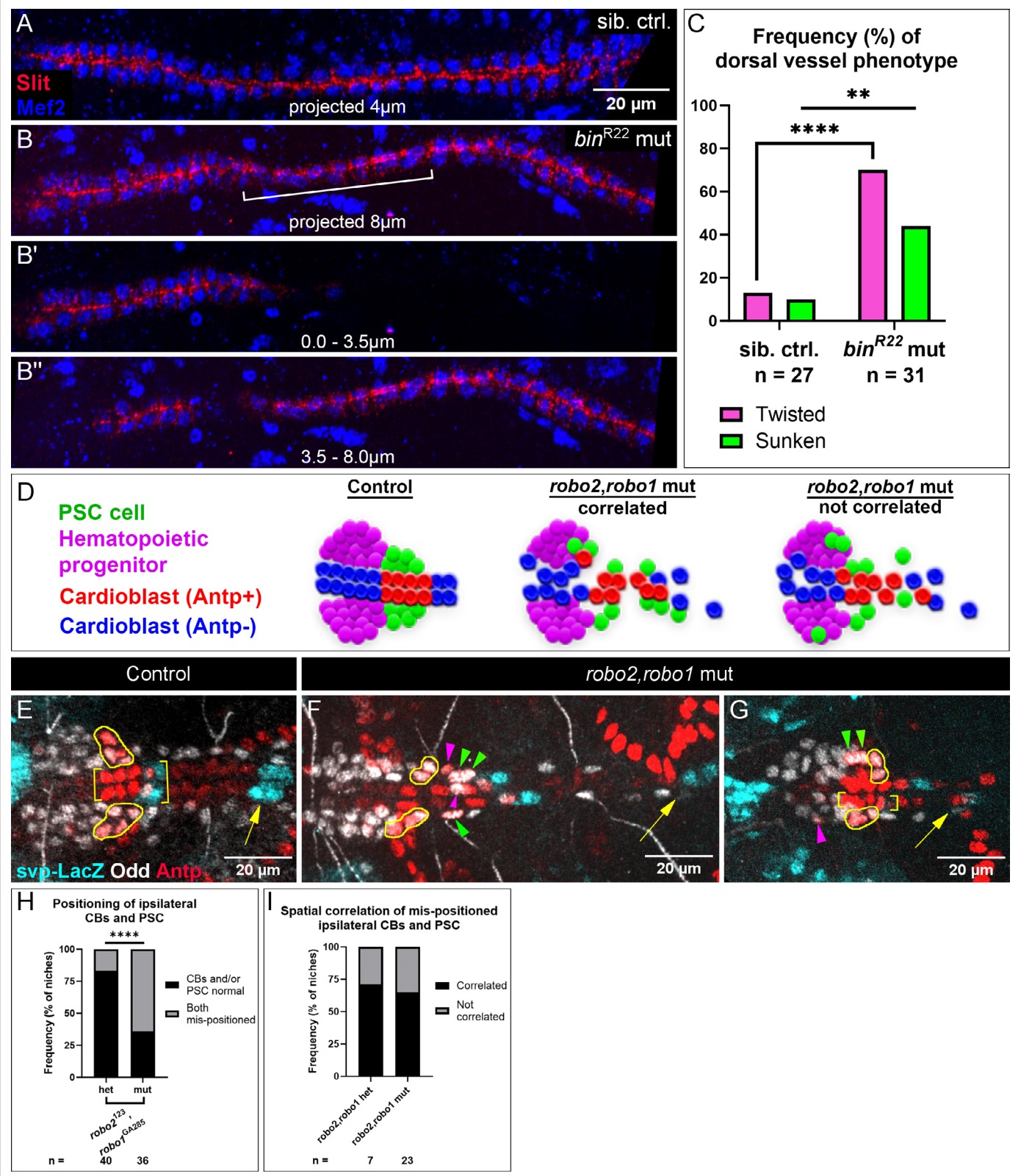

**Figure 4.** PSC positioning requires Vm-mediated dorsal vessel organization and Robo activation in PSC cells. (A-C) Analysis of dorsal vessels with and without Vm; phenotype frequencies quantified in (C). (A, B) Mef2-labeled CBs and Slit-labeled lumens of dorsal vessels. (A) Dorsal vessel of sibling control, captured in 4.0 μm projection, shows normal, neatly-aligned CB organization. (B) $bin^{R22}$ mutant with twisted dorsal vessel (bracket), fully captured only by projection of an 8.0 μm deep stack. (B′) twisted phenotype evidenced by the absence of the sunken portion in the first 3.5 μm;. (B″) 3.5–8 μm

*Figure 4 continued on next page*

*Figure 4 continued*

projection. (**D**) Schematics depicting three expectations for PSC and CB positioning. In controls (left), PSCs are coalesced and adjacent to CBs. In *robo2,robo1* mutants, either PSC cells are associated with mis-positioned CBs (middle), suggesting passive displacement of the PSCs; *or* displaced PSC cells are sometimes separate from mis-positioned CBs (right), suggesting the PSC cells themselves require activated Robo signaling. (**E-I**) Analysis of CB and PSC positioning with additional CB marker, *svp*-LacZ. Brackets indicate normally-positioned CBs. Yellow outlines indicate normally-positioned PSC cells. Arrowheads indicate abnormally-positioned PSC cells; green arrowheads indicate passive mis-positioning and magenta arrowheads are displaced PSC cells without a similarly displaced CB nearby. Arrows indicate the next set of svp +CBs to the posterior. (**E**) Control with normally-positioned PSCs and CBs; second set of svp +CBs (arrow) nearby. (**F**) *robo2,robo1* mutant with abnormal CB positioning; Antp +CBs are dispersed and the second set of svp +CBs (arrow) are displaced far posteriorly. PSCs display both passive mis-positioning posteriorly (green arrowheads) and mis-positioning without a similarly mispositioned CB nearby magenta arrowheads; bottom PSC cell invaded midline and top PSC cell is displaced laterally. (**G**) *robo2,robo1* mutant with left side CBs positioned normally (brackets) and the second set of svp +CBs in view (arrow); despite normal CB positioning, the ipsilateral PSC has a mis-positioned cell (magenta arrowhead). Right side CBs positioned abnormally (an Antp +CB is displaced laterally and the second set of svp +CBs are displaced posteriorly, not in view); correspondingly, the ipsilateral PSC has two laterally displaced cells (green arrowheads). (**H**) Frequency of mis-positioning of both a PSC and the ipsilateral CBs; this occurs more frequently in *robo2,robo1* mutants than heterozygotes. (**I**) Frequency of correlated mis-positioning of PSCs and CBs. Both *robo2,robo1* heterozygotes and mutants have CB-independent instances of PSC mis-positioning. Sample sizes as indicated. \*\*p<0.01, \*\*\*\*p<0.0001, Fisher's Exact test. Scale bars as indicated.

those instances of mis-positioning for whether the position of PSCs correlated with position of CBs. In about two-thirds of cases mis-positioning was correlated (*Figure 4F and G*, green arrowheads, quantified in 4I), suggesting that proper PSC positioning relies in part on proper CB positioning. However, in about one-third of cases PSC mis-positioning appeared independent of CB mis-positioning (*Figure 4F and G*, magenta arrowheads). We also observed PSC mis-positioning (*Figure 4G*, magenta arrowhead) when ipsilateral CBs were positioned normally (*Figure 4G*, brackets). These instances of non-correlation strongly suggest that Robo activation is required in PSC cells for their proper positioning.

Finally, we live-imaged *slit* mutants with Hand-RFP, Antp >myrGFP to examine the dynamics of PSC migration when signaling was compromised (*Figure 5—video 1*; *Figure 5—video 2*). In contrast to the robust collectivity noted throughout the migration of control PSCs (*Figure 1G*), this imaging revealed progressive deterioration of PSC integrity during migration. For instance, throughout imaging (*Figure 5—video 1*), one PSC cell (*Figure 5A*, cyan dot) bridged the gap between the Antp +CBs (*Figure 5A*, yellow brackets, 0 min) and posteriorly displaced CBs. Ultimately this PSC cell made aberrant contact with a CB from the contralateral side (*Figure 5A*, yellow dot, 70 min). This region had a persistent gap, possibly reflecting improper sealing of the dorsal vessel (*Figure 5A*, arrows, 112 min), as well as improper A-P alignment of Antp +CBs with their contralateral counterparts (*Figure 5A*, offset brackets, 112 min). Another PSC cell (*Figure 5A*, cyan arrowhead, 0–35 min) extended long protrusions along the peripheral edge of the PSC and was cleared by a macrophage (*Figure 5A*, yellow arrowhead, 42–49 min). At imaging onset in the same embryo (*Figure 5—video 2*), 2 PSC cells were laterally displaced (*Figure 5B*, arrowheads, 0–28 min) but connected to the main PSC collective by a thin protrusion (*Figure 5B*, green arrow, 0 min). These two cells detached from the collective (7–28 min), underwent cell shape changes (elongated at 42–56 min, brackets; highly protrusive at 77 min), and remained relatively stationary as the rest of the PSC and CBs migrated away (out of this focal plane).

Altogether, these data demonstrate that proper PSC formation requires (1) Slit signaling from the Vm and dorsal vessel where signaling is necessary for proper alignment and organization of the dorsal vessel, which, in turn, affects positioning of attached PSC cells; and (2) Robo activation in PSC cells for their persistent association with one another as a collective (*Figure 6*).

## Discussion

This is the first work to examine how the PSC is positioned. We describe the steady-state, coalesced positioning of the cells comprising the PSC, characterize their migration, and reveal the signaling requirements that facilitate their association and recruitment to the lymph gland. Altogether, the evidence we provide culminates in a model in which the PSC migrates to the dorsal midline as a collective of cells—some adhere to the dorsal vessel and the rest adhere to each other. We show that association of PSC cells requires input from Vm and dorsal vessel, and we implicate Slit as a necessary signal. Although we have not ruled out direct Vm-to-PSC signaling, we find at a minimum that Vm affects PSC positioning indirectly by its novel role in forming the dorsal vessel. The intricate regulation

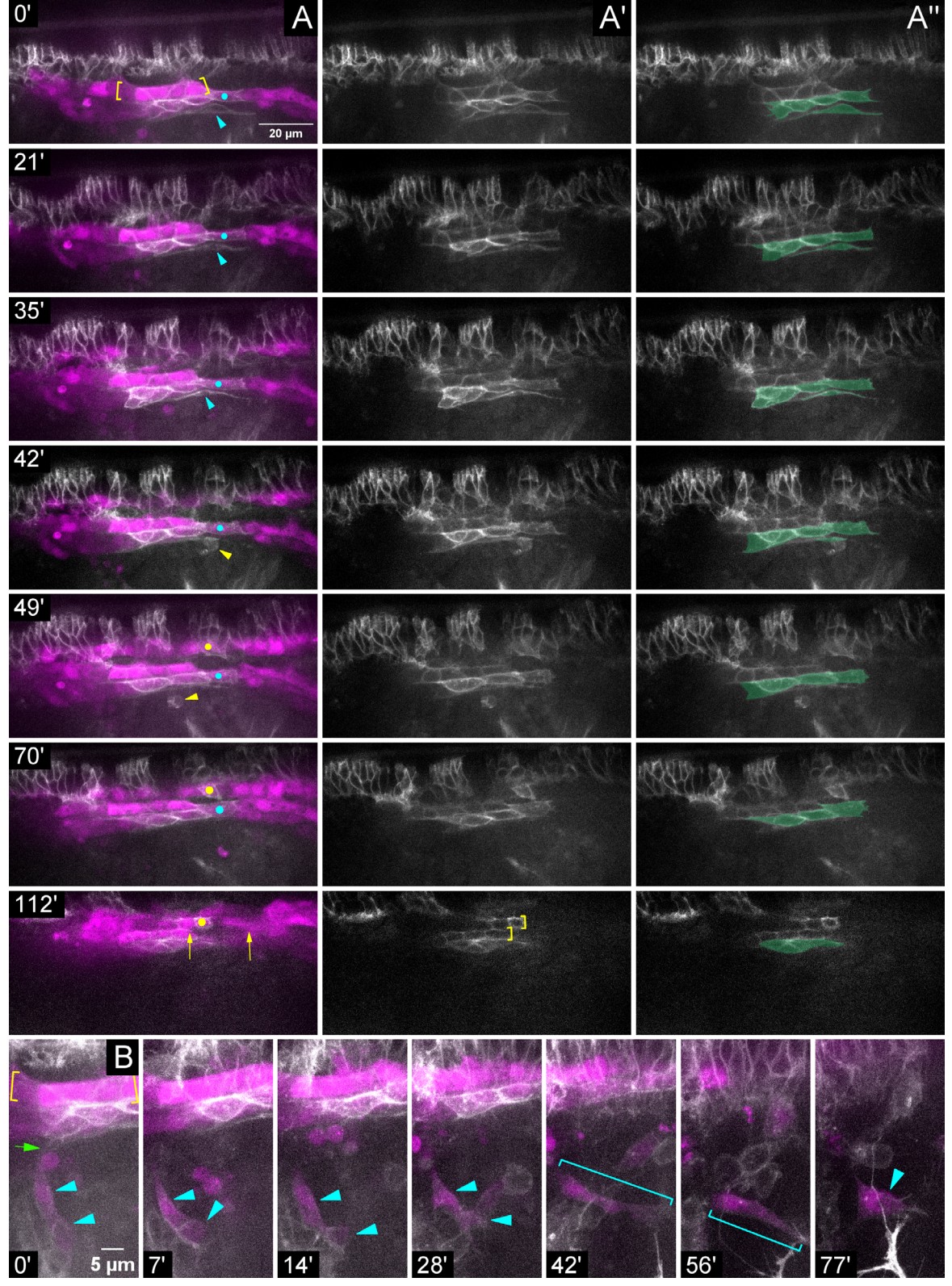

**Figure 5.** During migration Slit is required for proper PSC adhesions to CBs and for association of the PSC collective. Stills from timelapse imaging of *sli* mutant embryo with LG and PSC labeled by Hand-RFP, Antp >myrGFP; dorsolateral view. (**A**) Multichannel, (**A'**) single Antp >myrGFP channel, or (**A''**) single Antp >myrGFP channel false-colored for PSC cells; time increasing vertically (see timestamps). (**A**) The line of GFP+ cells within the brackets are Antp+ CBs; GFP+ cells below this are Antp+ PSC cells (see **A''**). Elongated PSC cell (cyan arrowhead; 0–35') cleared by macrophage (yellow arrowhead;

*Figure 5 continued on next page*

*Figure 5 continued*

42–49'). Anterior and posterior edges of another PSC cell (cyan dot) bridges separated CBs (0–70'); same PSC cell aberrantly contacts a contralateral Antp+ CB (yellow dot; 49–70'). Persisting gap in CBs (arrows; 112') evident in same region. (**A'**) Single Antp >myrGFP channel. Brackets indicate misaligned contralateral Antp+ CBs (112'). (**B**) Stills with time increasing across the row, revealing a different aspect of the same *sli* mutant embryo in (**A**). Cells within the yellow brackets are Antp+ CBs. The PSC contains two laterally displaced PSC cells (cyan arrowheads; 0–28') barely attached (green arrow; 0') to main cluster. These cells are disconnected from the main cluster (7') and remain stationary as the other PSC cells and CBs migrate away (7–77'). The disconnected PSC cell(s) change shape (cyan brackets) and develop membranous spikes (77' arrowhead). Scale bars as indicated.

The online version of this article includes the following video(s) for figure 5:

**Figure 5—video 1.** Live-imaging with dorsolateral view of PSC migration in sli2 mutant visualized with Hand-RFP, Antp>myrGFP (left) or Antp>myrGFP only (right) beginning midway in the migration.

https://elifesciences.org/articles/100455/figures#fig5video1

**Figure 5—video 2.** Live-imaging with dorsolateral view of PSC migration in sli2 mutant visualized with Hand-RFP, Antp>myrGFP.

https://elifesciences.org/articles/100455/figures#fig5video2

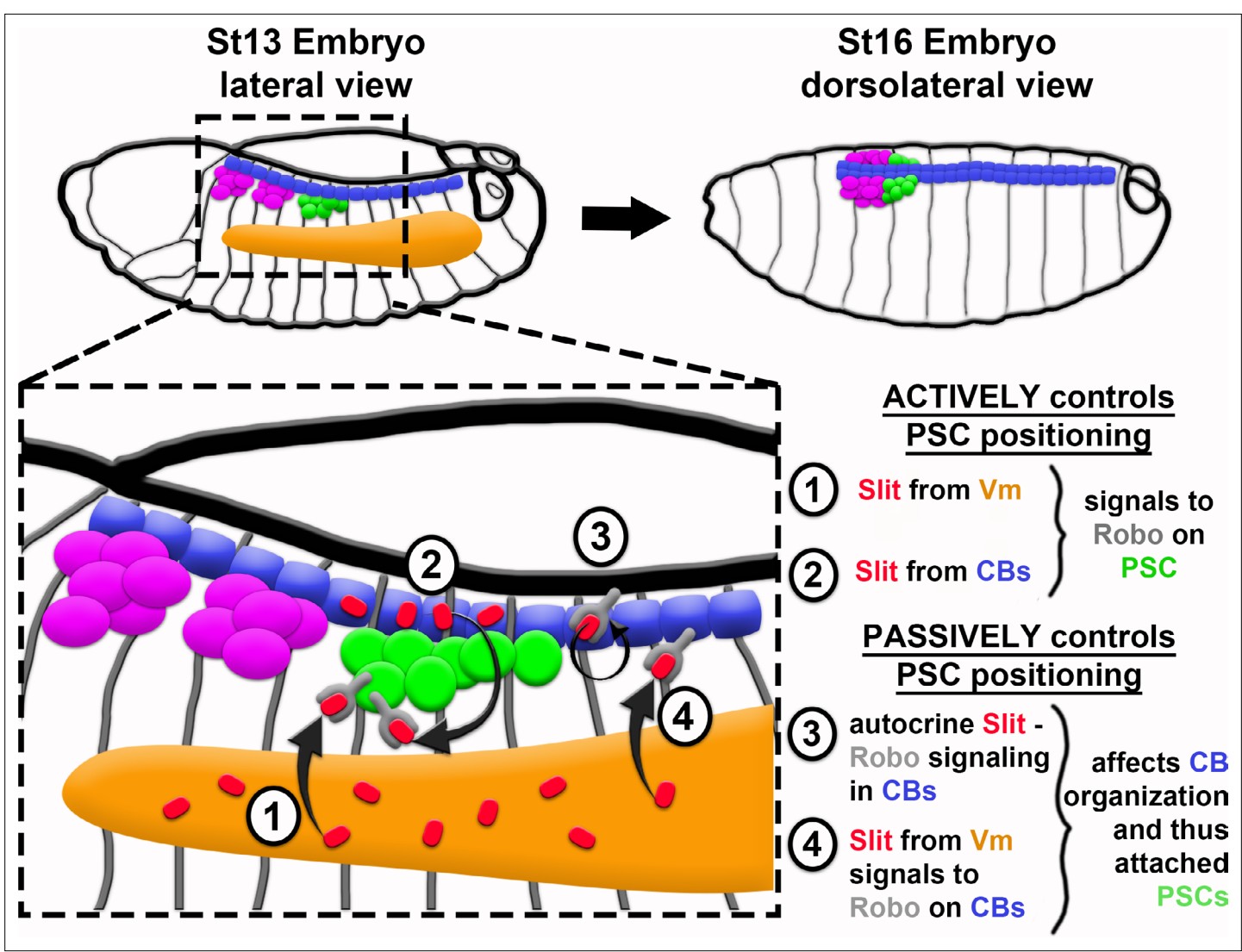

**Figure 6.** Model of PSC formation. PSC migration from its point of specification, laterally in the embryo (left), to its final position at the dorsal midline (right), requires both active Robo signaling in PSC cells (1 and 2) and proper organization of CBs – a passive, indirect control of PSC positioning (3 and 4). Slit from VM (1) and from CBs (2) directly impacts PSC positioning by binding to Robo1 and Robo2 receptors on PSC cells. In a more passive manner, Slit controls PSC positioning via binding to Robo receptors on CBs, which ensures their proper polarity and organization both by autocrine signaling in CBs (3; previously known) and by way of Slit emanating from Vm (4; novel finding from this work).

described herein ensures that PSC cells achieve a precise steady-state positioning as a coalesced group at the lymph gland posterior.

## Collective cell migration of the PSC

Prior knowledge of the embryonic PSC has been focused on PSC cell specification (*Crozatier et al., 2004*; *Mandal et al., 2007*). Thus, our live-imaging provides a substantial advancement by revealing PSC dynamics after specification, during migration. PSC formation is an example of collective cell migration, reminiscent of border cell migration in the *Drosophila* egg chamber and migration of the lateral line primordium in *Danio rerio*. These cell collectives respond to guidance cues, and their constituent cells are highly protrusive and shift position within the collective (*Cliffe et al., 2017*; *Dalle Nogare et al., 2020*; *Haas and Gilmour, 2006*; *Peercy and Starz-Gaiano, 2020*). PSC migration is no different (*Figure 1*)– we observe extension and retraction of protrusions on all surfaces of the collective except those in contact with CBs. This suggests that the migrating PSC explores and responds to its environment. Some cells of the PSC maintain their position adjacent to the dorsal vessel throughout the entirety of migration, while other cells exhibit fluidity within the collective. The positional shifts suggest remodeling of adhesive contacts between cells—albeit in a coordinated manner to achieve unidirectional migration for the entire cohort. Interestingly, circulating hemocytes can be seen interacting with the PSC—hemocytes are known to deposit and remodel ECM (*Bunt et al., 2010*), which could serve as a scaffold for adhesion molecules and as a substrate during migration. Indeed, we identified ECM components surrounding the PSC, including its interface with the dorsal vessel. The characteristics noted above could be further explored to improve our understanding not only of how the PSC is built, but more broadly of collective cell migration.

## PSC heterogeneity

For improved live-imaging, we sought a PSC-specific marker but found most candidates to be expressed in only subsets of PSC cells (unpublished data); the lack of homogenous expression suggests PSC heterogeneity. The existence of fixed and fluid PSC cell positionings in the collective further hints at heterogeneity—perhaps there are different types and polarities of adhesive molecules. If PSC heterogeneity exists at this early stage, its purpose is unknown. One possibility is that the fixed, dorsal vessel-adjacent PSC cells act as 'leaders' while the more fluid PSC cells are 'followers', a common phenomenon in collective cell migration (*Qin et al., 2021*). The dorsal vessel has recently been shown to serve as a second hematopoietic niche component (*Destalminil-Letourneau et al., 2021*; *Tian et al., 2023*), so another possibility is that the fixed, dorsal vessel-adjacent PSC cells have a special function as intermediaries that coordinate communication between the niche components. That said, whether the niche components interact remains unknown. Finally, PSC heterogeneity in this early context could reflect a division of labor later, when the PSC regulates larval hematopoietic progenitors. Gene expression heterogeneity has been noted in the testis niche (*Le Bras and Van Doren, 2006*; *Raz et al., 2023*; *Zheng et al., 2011*), suggesting that this may be a conserved feature of niches which warrants further investigation.

## Coordinated input of extrinsic signals positions the PSC

We reveal that positioning the PSC requires input from different extrinsic sources. It is perhaps notable that the few well-studied examples of niche formation all involve extrinsic inputs. The mammalian intestine is organized into protrusive villi and invaginated crypts; Wnt and EphB3 signals emitted from the crypt base recruit niche cells from the crypt-villus interface to their final position in the compact troughs of the crypts (*Batlle et al., 2002*; *Holmberg et al., 2006*; *van Es et al., 2005*). The *C. elegans* gonadal niche cell relies on adjacent germ cell proliferation for propulsion towards its steady-state position at the apex of the gonad (*Agarwal et al., 2022*). A microniche of the mammalian hair follicle, the dermal papilla, originates via extrinsic Fgf20 signaling to dermal fibroblasts. The fibroblasts are recruited into a condensate via directed migration, and then the condensate is segregated deeper into the skin via epidermal invagination driven by reciprocal signaling between dermis and epidermis (*Biggs et al., 2018*). The *Drosophila* testis niche cells require FGF and Slit signals from Vm to migrate through the testis and assemble as a cap that is further compacted via actomyosin contractility (*Anllo and DiNardo, 2022*; *Warder et al., 2024*). Together with our findings on PSC formation, a paradigm for niche formation is emerging that, subsequent to niche cell specification, extrinsic input segregates

niche cells from other constituents, and positions the collective toward one end of the tissue. Oftentimes positioning is accompanied by or precedes compaction of the recruited cells into a final niche shape/architecture.

We show that extrinsic Slit-Robo signaling is integral in establishing PSC position. Removing Slit causes PSC defects at the same frequency as removal of both canonical Robo receptors, indicating that in PSC formation, Slit only signals through Robo, and not through the non-canonical receptor, Dscam1. The PSC positioning phenotype was about 75% penetrant, and the severity of the defects varied. These observations argue that PSC formation relies on an additional cue(s) yet to be discovered. Evidence for an additional PSC coalescence cue exists in later larval stages, as manipulated Insulin Receptor signaling was shown to disrupt PSC coalescence (*Tokusumi et al., 2015*). Unfortunately, most pathways are difficult to test for a role in PSC formation due to lack of an embryonic PSC-specific tool with the necessary temporal control.

## Haploinsufficiency of Slit and Robo in PSC positioning

In the nervous system, one copy of the normal (WT) allele of Slit or Robo are typically sufficient for proper development (*Kidd et al., 1999*; *Rothberg et al., 1988*). In contrast, we find that about half of *robo2,robo1* double heterozygotes or of *slit* heterozygote embryos have abnormal PSCs, suggesting haploinsufficiency in this process. Our observation in embryonic stages is supported by previous findings at later larval stages where both *slit* or *robo2* heterozygotes have dispersed PSCs (*Morin-Poulard et al., 2016*). Perhaps haploinsufficiency in the PSC reflects different regulation and function of the Slit-Robo pathway from the nerve cord. Alternatively, because Slit-Robo signaling is required within both CBs and PSC for their proper development, the haploinsufficiency may reflect a combinatorial defect caused by diminished signaling in both tissues. Furthermore, we are surprised that *robo2* and *robo1* single mutants have a similar frequency of abnormal PSCs because in the embryo Robo1 is expressed in all PSC cells, whereas Robo2 appears to be expressed in only one. An intriguing possibility is that the Robo2-expressing PSC cell is the 'leader' in PSC migration, and without Robo2 signaling in the leader, the collective fails to migrate appropriately, or the followers fail to remain properly adherent. However, it may be that all PSC cells express Robo2, and our finding reflects limited detection ability with the Robo2-GFP reporter.

Morin-Poulard et. al. show that all larval PSC cells express Robo2, and it has a more prominent role than Robo1 in maintaining PSC coalescence (termed 'clustering' in their work) during larval stages (*Morin-Poulard et al., 2016*). In fact, they found Robo1 knockdown alone was insufficient to disrupt clustering. Taken together with our work, perhaps this indicates that Robo1 is more important for first establishing PSC coalescence and Robo2 is more important for maintaining it. Most manipulations occurred after the PSC had already formed, and therefore the changes reported cannot stem from defects during establishment but rather from dispersion of the cluster after it formed. One experiment without temperature control that could have diminished Robo signaling in PSC and CBs early (Antp >Robo RNAi), at about the time of PSC specification, generated a more dispersed late larval PSC compared to Robo knockdown after PSC formation. One can imagine that aberrant embryonic PSC positioning would become exacerbated and increasingly catastrophic as the PSC and nearby progenitors proliferate throughout larval stages.

## Function of Slit-Robo in PSC positioning

Determining how Slit affects a given process is complex because it can act as an attractive or repulsive cue, or even affect cell adhesion, and in some instances the outcome relies on whether Slit is cleaved by Tok (*Englund et al., 2002*; *Kellermeyer et al., 2020*; *Kolesnikov and Beckendorf, 2005*; *Kramer et al., 2001*). Thus, elucidating the role of the Slit-Robo pathway in PSC formation is difficult. The existence and relative positioning of two Slit sources, the reliance on one for positioning of the other, and the inconsistent direction of PSC mis-positioning suggest that it is too simplistic to ask whether Slit is acting as a repulsive or attractive cue for PSC cells. Notably, mis-positioned PSC cells adopted aberrant contacts with CBs, LG, and pericardial cells of the dorsal vessel. When reported in the dorsal vessel, this inappropriate mixing of cells was attributed to improper polarity of cell adhesion molecules (*Qian et al., 2005*; *Santiago-Martínez et al., 2006*). Furthermore, Slit is important for maintaining adhesion amongst larval PSC cells (*Morin-Poulard et al., 2016*). Taken together with our data, we reason that during PSC migration Slit must affect the ability of PSC cells to properly adhere

to the dorsal vessel and to one another. In this manner, Slit facilitates PSC cell association as a compact cluster.

The phenotypic variability we observe upon compromising Slit-Robo signaling indicates that downstream regulation is likely complex and dependent on the specific context, in time and space. *Morin-Poulard et al., 2016* show that the Slit-Robo pathway maintains clustering of larval PSC cells via DE-Cadherin and Cdc42; however, our data indicate different effectors are at play during PSC formation, as we detect no DE-Cad in the embryonic PSC (data not shown). Additionally, whereas constitutively active Cdc42 in the larval PSC caused dispersion (*Morin-Poulard et al., 2016*), our attempt to elicit PSC dispersion with this manipulation in the embryo yielded normally coalesced PSCs (data not shown). Therefore, we reason that different downstream effectors are engaged by the Slit-Robo pathway for initial positioning of the PSC.

This study was limited by the lack of a truly PSC-specific driver—a tool that would improve live-imaging and allow testing of potential Slit-Robo pathway effectors. We and others observe Fasciclin III, a homotypic cell adhesion molecule, in the PSC (unpublished data and *Mandal et al., 2007*), however, it is first detectable too late in embryogenesis to function during PSC formation. Thus, the particular adhesion molecules that maintain association of PSC cells as they migrate remain unknown. Elucidating this tool will facilitate interrogation of how Slit-Robo signaling impacts PSC adhesion to the dorsal vessel and amongst itself.

## Passive and active roles for Slit-Robo signaling in PSC positioning

Due to the necessity of autocrine Slit-Robo signaling in dorsal vessel formation (*Medioni et al., 2008*; *Qian et al., 2005*; *Santiago-Martínez et al., 2006*; *Santiago-Martínez et al., 2008*), it was challenging to distinguish whether effects on the PSC were secondary to mis-positioned CBs, or primary consequences of diminished Robo activation in PSC cells. Our analysis of *robo2,robo1* double mutants along with the registration marker, svp-lacZ, strongly suggests both passive mis-positioning of PSC cells via attachment to mis-positioned CBs and a requirement for Robo activation directly in PSC cells. A direct requirement for Robo in PSC cells is further suggested in *slit* mutant live-imaging. Here, the lateral-most PSC cells separated from the main cluster even though no CBs were nearby (*Figure 5B*). Perhaps during normal migration these more 'fluid', lateral PSC cells possess a stronger requirement for Robo activation and require direct input to achieve that. In this view, those PSC cells that are passively mis-positioned in mutants are the same PSC cells that remain affixed to the dorsal vessel during normal migration. Alternatively, or additionally, passive mis-positioning of CB-adjacent PSC cells might occur due to improper polarization of adhesive molecules, whereas separation of lateral PSC cells from the collective might occur due to a total loss of adhesive molecules. Finally, it is possible that PSC cells do not intrinsically require Robo activation, but rather CB-independent PSC mis-positioning in *sli* or *robo* mutants could be a secondary defect caused by compromised Slit-Robo signaling in some other tissue. A PSC-specific driver to knockdown Robo intrinsically would be needed to test definitively the requirement for Robo in PSC cells.

## Connecting niche structure, position, and function

The coalesced nature and posterior positioning of the PSC are its most prominent features. The consistency of these features suggested that they are under tight regulation, as we report here, and strongly indicates their relevance to function of the PSC. Although we have yet linked the precise architecture of the embryonic PSC to function, the specific architectures of multiple other niches have been shown to be functionally relevant. In the *Drosophila* gonad, a coalesced, compact niche is necessary for proper germline stem cell maintenance signaling, and for orienting stem cell divisions (*Anllo and DiNardo, 2022*; *Warder et al., 2024*). In the mammalian hair follicle, niche position determines stem cell fate (*Rompolas et al., 2013*). *Drosophila* neural stem cells are supported by glia, and the precise morphology of this niche is required for homeostasis of the entire nerve cord (*Spéder and Brand, 2018*). Most pertinently, recent work has shown that the larval PSC communicates amongst itself via a calcium signaling network. The integrity of this network is gap junction-dependent and is required for the PSC to emit proper levels of a progenitor maintenance signal (*Ho et al., 2021*; *Ho et al., 2023*). We find it reasonable to postulate that a dispersed PSC would exhibit defective calcium signaling such that the PSC would lack coordinated maintenance of progenitors, thereby leading to an imbalance in

the ratio of progenitors to differentiated hemocytes in the gland. Future experiments will determine how PSC coalescence is functionally relevant to its regulation of hematopoietic progenitors.

## Materials and methods

In this study, we used FlyBase (releases 2020_06–2024_02; *Jenkins et al., 2022*) to find information on phenotypes, function, stocks, and gene expression. All data describe biological replicates. Each experiment was repeated at least once. The Mann-Whitney test was used for statistical comparison of two groups of unpaired numerical data with non-Gaussian distribution. Fisher's exact test was used for statistical comparison of two groups of categorical data.

### *Drosophila* genetics

Detailed information on the *Drosophila* strains used in this study is in Appendix 1—key resources table. Controls were GAL4 only, a cross to $w^{1118}$, or a sibling control identified by a fluorescent balancer. This study generated the following embryo genotypes by combination into a stock (available upon email request) or obtained through a cross:

 Hand-RFP,Antp-GAL4,UAS-myr:GFP
 UAS-CD8:GFP / org-1-HN39-RFP; Antp-GAL4 / +
 UAS-CD8:GFP / +; tupAME-GAL4 / +
 UAS-CD8:GFP / UAS-grim; tupAME-GAL4 /TM6 Hu or MKRS
 $bin^{R22}$ / +
 $bin^{R22}$ / $bin^{S4}$
 bap-GAL4; + / CyODfd-YFP; bap-GAL4
 bap-GAL4; + / UAS-hid; bap-GAL4
 UAS-grim; tinCΔ4-GAL4
 $robo1^{GA285}$ /+
 $robo2^{1}$ /+
 $robo1^{GA285}$,$robo2^{123}$ /+
 tinCΔ4-GAL4 / +
 UAS-sli RNAi / +; tinCΔ4-GAL4/UAS-dcr-2
 UAS-dcr-2 / +; tinCΔ4-GAL4/UAS sli RNAi
 UAS-Robo1 OX / +; tinCΔ4-GAL4 / +
 bap-GAL4; + / UAS-Robo1 OX; bap-GAL4 /+
 $robo1^{GA285}$,$robo2^{123}$ /CyO,hindgut-LacZ; *svp*–LacZ / TM6Dfd-GFP
 $sli^{2}$; Hand-RFP,Antp-GAL4,UAS-myr:GFP
 $jeb^{weli}$ /+$jeb^{weli}$ / $jeb$[Df]

### Embryo collection

Unless otherwise noted, embryos were collected on apple juice agar plates overnight at 25°, unless an experiment involved a Gal4, in which case collections occurred at 29°. The next morning embryos were dechorionated in 50% bleach.

### Embryo fixation

Using a paintbrush, embryos were transferred from a collection basket to a 50/50 mixture by volume of heptane and 4% paraformaldehyde in Buffer B (16.7 mM $KPO_4$, pH 6.8; 75 mM KCl; 25 mM NaCl; 3.3 mM $MgCl_2$)(*de Cuevas and Spradling, 1998*). Embryos were fixed on a rocker for 15 min, then fixative was removed. An equal volume of MeOH was added and the vial was shaken vigorously for about 5 s to remove vitelline membranes. The heptane and MeOH mixture was discarded, and the embryos were rinsed three times with MeOH.

### Embryo staging and genotyping

Embryos were staged according to Atlas of *Drosophila* Development (*Hartenstein, 1993*). Embryos were genotyped according to presence of a balancer chromosome: CyO, P({Dfd-EYFP}); or CyO,P({Wg-lacZ}); or CyO, hindgut-lacZ; or TM6, P{Dfd-EYFP}, Sb, Hu, e.

## Immunostaining

Manipulations were room temperature unless otherwise noted. Embryos were rehydrated in 50%MeOH/50% PBS (10 mM $Na_2HPO_4$; 1.8 mM $KH_2PO_4$; 2.7 mM KCl; 137 mM NaCl; pH 7.4), followed by 100% PBS. Embryos rocked for 5 min in PBS with 0.1% Triton X-100 (PBST), then 1 hr in 4% normal donkey serum in PBST. Embryos were then transferred to a rocker at 4° until the end of the day or began incubation in primary antibody solution. The next day embryos were rinsed three times in PBST then rocked in PBST for 1 hr, incubated in secondary antibody solution for 1 hr, rinsed three times in PBST, rocked in PBST for 1 hr, equilibrated in 50% glycerol/50% Ringer's solution (5 mM HEPES, pH 7.3; 130 mM NaCl; 5 mM KCl; 2 mM $MgCl_2$; 2 mM $CaCl_2$) for 15 min or overnight at 4°, then mounted in 2% nPropyl-gallate in 90% glycerol. Primary antibodies were diluted in normal donkey serum as follows: mouse antibody against Antp (1:50; DSHB, 8C11), rabbit antibody against Odd skipped (1:400; gift from James Skeath, Washington University School of Medicine) chick antibody against GFP (1:1500; Aves Labs, GFP-1020), rabbit antibody against RFP (1:1000; Abcam, ab62341), mouse antibody against Fasciclin 3 (1:50; DSHB, 7G10), rabbit antibody against Mef2 (1:1000; DSHB), guinea pig antibody against Odd skipped (1:1200; gift from John Reinitz, University of Chicago), rabbit antibody against Bin (1:100; gift from Eileen Furlong, EMBL), mouse antibody against Slit (1:200; gift from Greg Bashaw, University of Pennsylvania), mouse antibody against Robo1 (1:200; gift from Greg Bashaw, University of Pennsylvania), chick antibody against LacZ (1:1000; Abcam, ab9361-250). Secondary antibodies (Alexa Fluor 488, Cy3, Cy5, and Alexa Fluor 647; Molecular Probes or Jackson ImmunoResearch) were all used at 3.75 μg/mL, for 1 hr.

## Live-imaging

After dechorionation, embryos were selected by stage and transgene expression, using a stereo-flourescent microscope, transferred to a piece of agar, and then oriented in a line with the ventral/ventrolateral surface facing up – hanging off the edge of the agar. A heptane-glue mixture was dried as a strip on a glass slide, and that surface touched to the embryos for transfer (now dorsal/dorso-lateral surface of embryos faces up). 3 μL of halocarbon oil was added atop the embryos. Bridging coverslips were glued to the slide on either side of the line of embryos, then the main coverslip was laid atop the bridging coverslips. We imaged every 5–10 min for 2–4.5 hr; Z-stacks spanned 25–40 μM with 0.3–1.0 μM step sizes.

## Microscopy

Fixed embryos were imaged on a Zeiss Axio Imager with ApoTome using a 40 x, 1.2 NA water immersion objective or a 20 x, 0.8 NA objective; z-steps were 0.5–1.0 uM. All live-imaging except for alary muscle imaging occurred with a CrestOptics X-Light V3 spinning disk confocal microscope using a 60 x, 1.3NA silicone immersion objective; images were captured with two pco.edge 4.2 bi sCMOS cameras operated by VisiView (Visitron) software. Alary muscles were live-imaged with an IX7 Olympus spinning disk confocal using a 63 x, NA 1.2 water immersion objective and captured with an EMCCD camera (Hamamatsu photonics, model C9100-13) controlled by MetaMorph software.

## PSC positioning phenotypic characterization

We used immunostains for the accepted markers, Antp and Odd, to identify PSC cells. Sometimes images contained Antp + and Odd + cells in epidermal stripes anterior to the LG; these were not considered PSC cells. An embryo was scored as having 'normal' PSC positioning if both PSCs were (1) coalesced within one nuclear diameter of one another, (2) adjacent to the dorsal vessel, and (3) at the same dorsal-ventral position as the posterior-most cells of the lymph gland. Abnormal PSC positioning presented as a range of phenotypes including PSCs dispersed into multiple groups, and PSCs that were coalesced but not located at the LG posterior. Fisher's Exact test was used for statistical analysis and $p < 0.05$ was considered statistically significant.

## AM ablation PSC cell counting

We considered a cell to be a PSC cell if it was co-labeled by Antp and Odd and was not located in epidermis. Number of PSC cells was recorded separately for the left and right PSCs of each embryo, and the total number of PSC cells per embryo was plotted. The Mann-Whitney test was used to determine a non-significant difference with $p > 0.05$.

## Bin fluorescent intensity

For 9 sibling controls and 10 Vm-ablated embryos, a normalized Bin fluorescent intensity was calculated for three different regions of st11 Vm founder cells with Bin-stained nuclei and Fas3-stained membranes. For st11 embryos imaged laterally, Fas3 labels the 2 lateral surfaces of a given Vm founder cell membrane. ImageJ software was used to extract fluorescent intensity values for regions of interest including large non-positive background regions in the embryo, founder cell nuclei, and founder cell membranes. Bin intensity for a given region was calculated by normalizing the background subtracted Bin level of a single representative nucleus to the average background subtracted Fas3 level of at least the two lateral surfaces of that same cell; most often an average Fas3 level for a given region was calculated based on the membranes of 2–4 founder cells. The normalized Bin fluorescent intensity for a given region is plotted. The Mann-Whitney test was used for statistical analysis to determine a significant difference of p<0.05.

## Tissue ablations

### AM ablation

The tupAME-GAL4 driver was combined with UAS-CD8:GFP for AM visualization, and GFP-labeled AMs were clearly visible in st13 controls. This line was crossed to UAS-grim to generate AM-ablated embryos; AMs were ablated by st14.

### CB ablation

Flies with the tinCΔ4-GAL4 driver were crossed to flies with either UAS-grim or UAS-hid and embryos were collected overnight at 29°. Expression of either pro-apoptotic gene led to quite effective dorsal vessel ablation, but there were also defects in either Vm (for Grim) or in germband retraction (for Hid; data not shown), which could confound a PSC positioning analysis. We circumvented this issue by lowering GAL4 activity—embryos from the cross to UAS-grim were collected at 25°—which yielded relatively normal Vm but successfully ablated CBs beginning at st14.

## Slit RNAi in dorsal vessel

Two independent UAS-controlled RNAi's targeting different regions of Exon 4 of Slit along with UAS-controlled Dcr2 were driven by tinCΔ4-GAL4. Knockdown was evident at st14, the same stage when Slit expression becomes discernable in control CBs.

## Acknowledgements

We thank the Bloomington *Drosophila* Stock Center (NIH P40OD018537), J Skeath, J Reinitz, E Furlong, G Bashaw, M Crozatier, U Banerjee, M Frasch, G Vogler, and DSHB for antibodies and stocks. We thank CDB microscopy core director Andrea Stout for advice on imaging. Thanks to B Warder, G Vida, G Bashaw, and the K Lenhart lab for input during this project. This work was supported by NIH grants T32 HD083185 and F31 HD111208 to KAN; R01 GM138705 to KFL; F32 GM125123 to LA; R35 GM136270 and R03 HD111973 to SD.

## Additional information

### Funding

| Funder | Grant reference number | Author |
|---|---|---|
| Eunice Kennedy Shriver National Institute of Child Health and Human Development | HD083185 | Kara A Nelson |
| Eunice Kennedy Shriver National Institute of Child Health and Human Development | HD111208 | Kara A Nelson |

| Funder | Grant reference number | Author |
| --- | --- | --- |
| National Institute of General Medical Sciences | GM138705 | Kari F Lenhart |
| National Institute of General Medical Sciences | GM125123 | Lauren Anllo |
| National Institute of General Medical Sciences | GM136270 | Stephen DiNardo |
| Eunice Kennedy Shriver National Institute of Child Health and Human Development | HD111973 | Stephen DiNardo |

The funders had no role in study design, data collection and interpretation, or the decision to submit the work for publication.

## Author contributions

Kara A Nelson, Conceptualization, Resources, Formal analysis, Funding acquisition, Investigation, Visualization, Methodology, Writing – original draft, Writing – review and editing; Kari F Lenhart, Visualization, Writing – review and editing; Lauren Anllo, Supervision; Stephen DiNardo, Conceptualization, Supervision, Funding acquisition, Validation, Methodology, Project administration, Writing – review and editing

## Author ORCIDs

Kara A Nelson https://orcid.org/0000-0003-4847-2835
Lauren Anllo https://orcid.org/0000-0001-5482-5882
Stephen DiNardo https://orcid.org/0000-0003-4131-5511

Reviewer #1 (Public review): https://doi.org/10.7554/eLife.100455.3.sa1
Reviewer #2 (Public review): https://doi.org/10.7554/eLife.100455.3.sa2
Reviewer #3 (Public review): https://doi.org/10.7554/eLife.100455.3.sa3
Author response https://doi.org/10.7554/eLife.100455.3.sa4

---

# Additional files

## Supplementary files

MDAR checklist

## Data availability

All data generated or analysed during this study are included in the manuscript and supporting files.

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

# Appendix 1

**Appendix 1—key resources table**

| Reagent type (species) or resource | Designation | Source or reference | Identifiers | Additional information |
|---|---|---|---|---|
| Genetic reagent (*D. melanogaster*) | Antp-GAL4 | *Emerald and Cohen, 2004* | FLYB:FBal0155891 | FlyBase symbol: GAL4[Antp-21] |
| Genetic reagent (*D. melanogaster*) | w[1118] | Bloomington *Drosophila* Stock Center | BDSC:3605; FLYB:FBal0018186; RRID:BDSC_3605 | FlyBase symbol: w[1118] |
| Genetic reagent (*D. melanogaster*) | Hand-RFP | other | | Gift from Georg Vogler |
| Genetic reagent (*D. melanogaster*) | UAS-myr:GFP | Bloomington *Drosophila* Stock Center | BDSC:32200; FLYB:FBti0131976 RRID:BDSC_32200 | FlyBase symbol: P{10XUAS-IVS-myr::GFP} su(Hw)attP1 |
| Genetic reagent (*D. melanogaster*) | tupAME-GAL4 | *Bataillé et al., 2020* | FLYB: FBtp0142468 | FlyBase symbol: P{tup-GAL4.AME-R} Gift from J.L. Frendo |
| Genetic reagent (*D. melanogaster*) | UAS-CD8:GFP | other | | Gift from J.L. Frendo |
| Genetic reagent (*D. melanogaster*) | org-1-HN39-RFP | *Schaub et al., 2015* | FLYB:FBal0276776 | FlyBase symbol: RFP[org-1.HN39] |
| Genetic reagent (*D. melanogaster*) | UAS-grim | Hugo Bellen | FLYB:FBti0154788 | Flybase symbol: Dmel\P{UAS-grim.Y}2 |
| Genetic reagent (*D. melanogaster*) | bin[R22] | *Zaffran et al., 2001* | FLYB:FBal0043738 | Flybase symbol: Dmel\bin[R22] |
| Genetic reagent (*D. melanogaster*) | bin[S4] | *Zaffran et al., 2001* | FLYB:FBal0043739 | Flybase symbol: Dmel\bin[S4] |
| Genetic reagent (*D. melanogaster*) | UAS-hid | Bloomington *Drosophila* Stock Center | BDSC:65403; FLYB:FBti0183136 RRID:BDSC_65403 | Flybase symbol: Dmel\P{UAS-hid.Z}2 |
| Genetic reagent (*D. melanogaster*) | bap-GAL4 | *Zaffran et al., 2001* | BDSC:91540; FLYB:FBti0214156 | Flybase symbol: Dmel\P{bap-GAL4.3}1.1 |
| Genetic reagent (*D. melanogaster*) | bap-GAL4 | other | | gift from Manfred Frasch; Chr: X |
| Genetic reagent (*D. melanogaster*) | tinCΔ4-GAL4 | Bloomington *Drosophila* Stock Center | BDSC:92965; FLYB:FBti0216630 | Flybase symbol: Dmel\P{tinC-Gal4.Δ4}12 a |
| Genetic reagent (*D. melanogaster*) | slit[2] | Bloomington *Drosophila* Stock Center | BDSC:3266 FLYB:FBal0015700 | Flybase symbol: Dmel\sli[2] |
| Genetic reagent (*D. melanogaster*) | robo1[GA285] | other | FLYB:FBal0032588 | Gift from Greg Bashaw Flybase symbol: Dmel\robo1[1] |
| Genetic reagent (*D. melanogaster*) | robo2[1] | *Rajagopalan et al., 2000* | FLYB:FBal0121562 | Gift from Greg Bashaw Flybase symbol: Dmel\robo2[1] |
| Genetic reagent (*D. melanogaster*) | robo2[123] | other | FLYB:FBal0123720 | Gift from Greg Bashaw Flybase symbol: Dmel\robo2[X123] |
| Genetic reagent (*D. melanogaster*) | UAS-Slit RNAi #1 | Vienna *Drosophila* Stock Center | VDRC:v108853 FLYB:FBti0159991 | Flybase symbol: Dmel\P{KK100803}VIE-260B |
| Genetic reagent (*D. melanogaster*) | UAS-Slit RNAi #2 | Bloomington *Drosophila* Stock Center | BDSC:31468 FLYB:FBal0245521 | Flybase symbol: Dmel\sli[JF01229] |
| Genetic reagent (*D. melanogaster*) | UAS-Robo1 OX | *Evans et al., 2015* | BDSC:97240 FLYB:FBal0316479 | Flybase symbol: Dmel\robo1 ΔC.10xUAS. Tag:HA,Tag:SS(wg) |
| Genetic reagent (*D. melanogaster*) | UAS-dcr2 | Bloomington *Drosophila* Stock Center | BDSC:24650 FLYB:FBti0100275 | Flybase symbol: Dmel\P{UAS-Dcr-2.D}2 |

*Appendix 1 Continued on next page*

*Appendix 1 Continued*

| Reagent type (species) or resource | Designation | Source or reference | Identifiers | Additional information |
|---|---|---|---|---|
| Genetic reagent (*D. melanogaster*) | UAS-dcr2 | Bloomington *Drosophila* Stock Center | BDSC:24651 FLYB:FBti0100276 | Flybase symbol: Dmel\P{UAS-Dcr-2.D}10 |
| Genetic reagent (*D. melanogaster*) | svp-lacZ | Bloomington *Drosophila* Stock Center | BDSC:7314 FLYB:FBti0002862 | Flybase symbol: Dmel\P{HZ}svp$^3$ |
| Genetic reagent (*D. melanogaster*) | perlecan-GFP | Flytrap; GFP Protein Trap Database | FLYB:FBal0243609 | Flybase symbol: Dmel\trol$^{ZCL1700}$ |
| Genetic reagent (*D. melanogaster*) | viking-GFP | *Buszczak et al., 2007* | FLYB:FBal0211825 | Flybase symbol: Dmel\vkg$^{CC00791}$ |
| Genetic reagent (*D. melanogaster*) | *bap*$^{208}$ | Bloomington *Drosophila* Stock Center | BDSC:91539 FLYB:FBal0034201 | Flybase symbol: Dmel\bap$^{208}$ |
| Genetic reagent (*D. melanogaster*) | *jeb*$^{weli}$ | *Stute et al., 2004* | FLYB:FBal0159133 | Flybase symbol: Dmel\jeb$^{weli}$ |
| Genetic reagent (*D. melanogaster*) | *jeb* Df | Bloomington *Drosophila* Stock Center | BDSC:26551 FLYB:FBab0045764 | Flybase symbol: Df(2 R)BSC699 |
| Genetic reagent (*D. melanogaster*) | robo2-GFP | Bloomington *Drosophila* Stock Center | BDSC:61774 FLYB:FBal0265307 | Flybase symbol: Dmel\robo2$^{MI04295}$ |
| Antibody | anti-Antp (Mouse monoclonal) | Developmental Studies Hybridoma Bank | Cat#:8C11, RRID:AB_528083 | IF(1:50) |
| Antibody | anti-Odd skipped (Rabbit polyclonal) | *Ward and Skeath, 2000* | | IF(1:400); gift from James Skeath |
| Antibody | anti-GFP (Chick polyclonal) | Aves labs | Cat#:GFP-1020 RRID:AB_2307313 | IF(1:1500) |
| Antibody | anti-Fas3 (Mouse monoclonal) | Developmental Studies Hybridoma Bank | Cat#:7G10 RRID:AB_528238 | IF(1:50) |
| Antibody | anti-Mef2 (Rabbit polyclonal) | Developmental Studies Hybridoma Bank | Cat#:Mef2 RRID:AB_2892602 | IF(1:1000) |
| Antibody | anti-Slit (Mouse monoclonal) | Developmental Studies Hybridoma Bank | Cat#:C555.6D RRID:AB_528470 | IF(1:200); gift from Greg Bashaw |
| Antibody | anti-LacZ (Chick polyclonal) | Abcam | Cat#:ab9361 RRID:AB_307210 | IF(1:1000) |
| Antibody | anti-RFP (Rabbit polyclonal) | Abcam | Cat#:ab62341 RRID:AB_945213 | IF(1:1000) |
| Antibody | anti-Bin (Rabbit polyclonal) | other | | IF(1:100); gift from Eileen Furlong |
| Antibody | anti-Robo1 (Mouse monoclonal) | other | | IF(1:200); gift from Greg Bashaw |
| Antibody | anti-Odd skipped (Guinea pig polyclonal) | other | | IF(1:1200); gift from John Reinitz |
| Chemical compound, drug | Paraformaldehyde | Electron Microscopy Sciences | Cat#:15710 | |
| Chemical compound, drug | Propyl-gallate | Sigma Aldrich | PubChem Substance ID:24898394; SKU:P3130; CAS Number:121-79-9 | |
| Chemical compound, drug | Normal Donkey Serum | Jackson ImmunoResearch Labs Inc | Cat#:017-000-121 RRID:AB_2337258 | |
| Chemical compound, drug | Ringer's solution | other | | Recipe from *de Cuevas and Spradling, 1998* |

*Appendix 1 Continued on next page*

*Appendix 1 Continued*

| Reagent type (species) or resource | Designation | Source or reference | Identifiers | Additional information |
|---|---|---|---|---|
| Chemical compound, drug | Triton X-100 | MilliporeSigma | CAS Number: 9036-19-5 | |
| Software, algorithm | FIJI | ImageJ | RRID:SCR_002285 | http://fiji.sc |
| Software, algorithm | Photoshop | Adobe | RRID:SCR_014199 | https://www.adobe.com/products/photoshop.html |
| Software, algorithm | Prism | Graphpad | RRID:SCR_002798 | v9.0.0-v10.0.0 |
| Software, algorithm | Axio-Vision Imaging Software | Zeiss | | v4.8.1 |
| Software, algorithm | VisiView | Visitron | | |
| Software, algorithm | Metamorph Microscopy Automation and Image Analysis Software | Leica | | v7.8.40 |
| Other | 63 x / 1.2 NA water immersion objective | Leica | | |
| Other | 60 x / 1.3 NA silicone immersion objective | Olympus | | |
| Other | AxioCam HRm | Zeiss | | |
| Other | 40 x / 1.2 NA water immersion objective | Zeiss | | |
| Other | 20 x / 0.8 NA objective | Zeiss | | |
| Other | M165FC | Leica | | |
| Other | Achromat 1.6 x objective | Leica | | |
| Other | GFP Filter set ET470/40 x; ET525/50 m | Leica | | |
| Other | mCherry Filter set ET560/40 x; ET630/75 m | Leica | | |
| Other | pco.edge 4.2 bi sCMOS | PCO | | |
| Other | Cell Center Stockroom (Penn) | other | RRID:SCR_022399 | |
| Other | CDB Microscopy Core (Penn) | other | RRID:SCR_022373 | |

