## [Editor Report · eLife Assessment]

This study presents **valuable** findings on the role of a well-studied signal transduction pathway, the Slit/Robo system, in the context of the assembly of the hematopoietic niche in the *Drosophila* embryo. The evidence supporting the claims of the authors is **solid**. The work will interest developmental biologists working on molecular mechanisms of tissue morphogenesis.

---

## [Referee Report · Reviewer #1 (Public review)]

Summary:

The study by Nelson et al. is focused on formation of the *Drosophila* Posterior Signaling Center (PSC) which ultimately acts as a niche to support hematopoietic stem cells of the lymph gland (LG). Using a combination of genetics and live imaging, the authors show that PSC cells migrate as a tight collective and associate with multiple tissues during a trajectory that positions them at the posterior of the LG.

This is an important study that identifies Slit-Robo signaling as a regulator of PSC morphogenesis, and highlights the complex relationship of interacting cell types - PSC, visceral mesoderm (VM) and cardioblasts (CBs) - in coordinated development of these three tissues during organ development. However, one point requiring clarification is the idea that PSC cells exhibit a collective cell migration; it is not clear that the cells are migrating rather than being pushed to a more dorsal position through dorsal closure and/or other similar large scale embryo movement. This does not detract from the very interesting analysis of PSC morphogenesis as presented.

Strengths:

• Using expression of Hid or Grim to ablate associated tissues, they find evidence that the VM and CB of the dorsal vessel affect PSC migration/morphology whereas the alary muscles do not. Slit is expressed by both VM and CBs, and therefore Slit-Robo signaling was investigated as PSCs express Robo.

• Using a combination of approaches, the authors convincingly demonstrate that Slit expression in the CBs and VM acts to support PSC positioning. A strength is the ability to knockdown slit levels in particular tissue types using the Gal4 system and RNAi.

• Although in the analysis of robo mutants, the PSC positioning phenotype is weaker in the individual mutants (robo1 and robo2) with only the double mutant (robo1,robo2) exhibiting a phenotype comparable to the slit RNAi. The authors make a reasonable argument that Slit-Robo signaling has an intrinsic effect, likely acting within PSCs, because PSCs show a phenotype even when CBs do not (Fig 4G).

• New insight into dorsal vessel formation by VM is presented in Fig 4A,B, as loss of the VM can affect dorsal vessel morphogenesis. This result additionally points to the VM as important.

Weaknesses:

• The authors are cautioned to temper the result that Slit-Robo signaling is intrinsic to PSC since loss of robo may affect other cell types (besides CBs and PSCs) to indirectly affect PSC migration/morphogenesis. In fact, in the robo2, robo1 mutant, the VM appears to be incorrectly positioned (Fig. 4G).

• If possible, the authors should use RNAi to knockdown Robo1 and Robo2 levels specifically in the PSCs if a Gal4 is available; might Antp.Gal4 (Fig 1K) be useful? Even if knockdown is achieved in PSCs+CBs, this would be a better/complementary experiment to support the approach outlined in Fig 4D.

• Movies are hard to interpret, as it seems unclear that the PSCs actively migrate rather than being pushed/moved indirectly due to association with VM and CBs/dorsal vessel.

---

## [Referee Report · Reviewer #2 (Public review)]

The paper by Nelson KA, et al. explored the collective migration, coalescence and positioning of the posterior signaling center (PSC) cells in *Drosophila* embryo. With live imaging, the authors observed the dynamic progress of PSC migration. Throughout this process, visceral mesoderm (VM), alary muscles (Ams) and cardioblasts (CBs) are in proximity of PSC. Genetic ablation of these tissues reveals the requirement for VM and CBs, but not AMs in this process. Genetic manipulations further demonstrated that Slit-Robo signaling was critical during PSC migration and positioning. While the genetic mechanisms of positioning the PSC were explored in much detail, including using live imaging, the functional consequence of mispositioning or (partial) absence of PSC cells has not been addressed, but would much increase the relevance of their findings. A few additional issues need to be addressed as well in this otherwise well-done study.

Previous major points:

(1) The only readout in their experiments is the relative correctness of PSC positioning. Importantly, what is the functional consequence if PSC is not properly positioned? This would be particularly important with robo-sli manipulations, where the PSC is present but some cells are misplaced. What is the consequence? Are the LGs affected, like specification of their cell types, structure and function? To address this for at least the robo-slit requirement in the PSC, it may be important to manipulate them directly in the PSC with a split Gal4 system, using Antp and Odd promoters.

(2) The densely, parallel aligned fibers in the lower part of Figure 1J seemed to be visceral mesoderm, but further up (dorsally) that may be epidermis. It is possible that the PSC migrate together with the epidermis? This should be addressed.

(3) Although the authors described the standards of assessing PSC positioning as "normal" or "abnormal", it is rather subtle at times and variable in the mutant or KD/OE examples. The criteria should be more clearly delineated and analyzed double-blind, also since this is the only readout. Further examples of abnormal positioning in supplementary figures would also help.

(4) Discussion is very lengthy and should shortened.

Comments on revised version:

Although the authors have responded to my concerns as they deemed suitable, these concerns still stand for the revised version.

---

## [Referee Report · Reviewer #3 (Public review)]

Summary:

This work is a detailed and thorough analysis of the morphogenesis of the posterior signaling center (PSC), a hematopoietic niche in the *Drosophila* larva. Live imaging is performed from the stage of PSC determination until the appearance of a compact lymph gland and PSC in the stage 16 embryo. This analysis is combined with genetic studies that clarify the involvement of adjacent tissue, including the visceral mesoderm, alary muscle, and cardioblasts/dorsal vessel. Lastly, the Slit/Robo signaling system is clearly implicated in the normal formation of the PSC.

Strengths:

The data are clearly presented and well documented, and fully support the conclusions drawn from the different experiments.

The authors have addressed all of my previous comments, in particular concerning the role of epidermal cell rearrangements during dorsal closure as a possible force acting on the movement of PSC cells. The authors have clarified their definition of "collective migration" as it applies to the movement of PSC. The revised paper will make an important contribution to our understanding of the mechanisms driving morphogenesis.

---

## [Author Response]

The following is the authors’ response to the current reviews.

**Reviewer #1 (Public review):**
Summary:The study by Nelson et al. is focused on formation of the *Drosophila* Posterior Signaling Center (PSC) which ultimately acts as a niche to support hematopoietic stem cells of the lymph gland (LG). Using a combination of genetics and live imaging, the authors show that PSC cells migrate as a tight collective and associate with multiple tissues during a trajectory that positions them at the posterior of the LG.This is an important study that identifies Slit-Robo signaling as a regulator of PSC morphogenesis, and highlights the complex relationship of interacting cell types - PSC, visceral mesoderm (VM) and cardioblasts (CBs) - in coordinated development of these three tissues during organ development. However, one point requiring clarification is the idea that PSC cells exhibit a collective cell migration; it is not clear that the cells are migrating rather than being pushed to a more dorsal position through dorsal closure and/or other similar large scale embryo movement. This does not detract from the very interesting analysis of PSC morphogenesis as presented.

This Public Review by Reviewer #1 is identical to their original Public Review, thus we are unsure whether Reviewer #1 assessed the revised version of our manuscript, and whether they read our responses to their original Public Review. Below we summarize our original responses to the weaknesses listed for the first version of our manuscript.

Strengths:• Using expression of Hid or Grim to ablate associated tissues, they find evidence that the VM and CB of the dorsal vessel affect PSC migration/morphology whereas the alary muscles do not. Slit is expressed by both VM and CBs, and therefore Slit-Robo signaling was investigated as PSCs express Robo.• Using a combination of approaches, the authors convincingly demonstrate that Slit expression in the CBs and VM acts to support PSC positioning. A strength is the ability to knockdown slit levels in particular tissue types using the Gal4 system and RNAi.• Although in the analysis of robo mutants, the PSC positioning phenotype is weaker in the individual mutants (robo1 and robo2) with only the double mutant (robo1,robo2) exhibiting a phenotype comparable to the slit RNAi. The authors make a reasonable argument that Slit-Robo signaling has an intrinsic effect, likely acting within PSCs, because PSCs show a phenotype even when CBs do not (Fig 4G).• New insight into dorsal vessel formation by VM is presented in Fig 4A,B, as loss of the VM can affect dorsal vessel morphogenesis. This result additionally points to the VM as important.Weaknesses:• The authors are cautioned to temper the result that Slit-Robo signaling is intrinsic to PSC since loss of robo may affect other cell types (besides CBs and PSCs) to indirectly affect PSC migration/morphogenesis. In fact, in the robo2, robo1 mutant, the VM appears to be incorrectly positioned (Fig. 4G).

We maintain our conclusion, and, we point out that the Reviewer stated, “The authors make a reasonable argument that Slit-Robo signaling has an intrinsic effect, likely acting within PSCs”. We already added a statement to the Discussion reminding the reader of the possibility of secondary defects (“Finally, it is possible that PSC cells do not intrinsically require Robo activation, but rather CB-independent PSC mis-positioning in *sli* or *robo* mutants could be a secondary defect caused by compromised Slit-Robo signaling in some other tissue.”).

• If possible, the authors should use RNAi to knockdown Robo1 and Robo2 levels specifically in the PSCs if a Gal4 is available; might Antp.Gal4 (Fig 1K) be useful? Even if knockdown is achieved in PSCs+CBs, this would be a better/complementary experiment to support the approach outlined in Fig 4D.

As described in our first response, use of Antp-GAL4 with RNAi would be no better than a whole animal double Robo mutant.

• Movies are hard to interpret, as it seems unclear that the PSCs actively migrate rather than being pushed/moved indirectly due to association with VM and CBs/dorsal vessel.

Vm does not directly contact the PSC, so the Vm cannot be physically pushing the PSC. In their original review, Reviewer #3 expressed similar concerns (Weaknesses #1 and #2), and upon their review of our revised manuscript they determined we addressed these concerns.

**Reviewer #2 (Public review):**
The paper by Nelson KA, et al. explored the collective migration, coalescence and positioning of the posterior signaling center (PSC) cells in *Drosophila* embryo. With live imaging, the authors observed the dynamic progress of PSC migration. Throughout this process, visceral mesoderm (VM), alary muscles (Ams) and cardioblasts (CBs) are in proximity of PSC. Genetic ablation of these tissues reveals the requirement for VM and CBs, but not AMs in this process. Genetic manipulations further demonstrated that Slit-Robo signaling was critical during PSC migration and positioning. While the genetic mechanisms of positioning the PSC were explored in much detail, including using live imaging, the functional consequence of mispositioning or (partial) absence of PSC cells has not been addressed, but would much increase the relevance of their findings. A few additional issues need to be addressed as well in this otherwise well-done study.Previous major points:(1) The only readout in their experiments is the relative correctness of PSC positioning. Importantly, what is the functional consequence if PSC is not properly positioned? This would be particularly important with robo-sli manipulations, where the PSC is present but some cells are misplaced. What is the consequence? Are the LGs affected, like specification of their cell types, structure and function? To address this for at least the robo-slit requirement in the PSC, it may be important to manipulate them directly in the PSC with a split Gal4 system, using Antp and Odd promoters.

We state in our original response that exploring the functional consequences of PSC mis-positioning was outside the scope of this study. Given that the necessary cis-regulatory modules have not been identified at Antp or Odd, creating a split-GAL4 with ‘Antp and Odd promoters’ cannot be accomplished in a reasonable time frame, as we previously detailed in our original response.

(2) The densely, parallel aligned fibers in the lower part of Figure 1J seemed to be visceral mesoderm, but further up (dorsally) that may be epidermis. It is possible that the PSC migrate together with the epidermis? This should be addressed.

This was directly addressed by the additional data included in our revision. When epidermal closure is stalled, the PSC is able to migrate past the stalled leading edge, closer to the midline.

(3) Although the authors described the standards of assessing PSC positioning as "normal" or "abnormal", it is rather subtle at times and variable in the mutant or KD/OE examples. The criteria should be more clearly delineated and analyzed double-blind, also since this is the only readout. Further examples of abnormal positioning in supplementary figures would also help.

We addressed this comment in detail in our original response. Briefly, double-blinding was oftentimes not possible due to the obviousness of the genotype in the image. The criteria we outline for normal PSC positioning is as comprehensive as possible given the subtlety variability of mis-positioning phenotypes. Two of the authors independently analyzed the relatively large sets of samples and arrived at the same conclusions.

(4) Discussion is very lengthy and should shortened.

We shortened the Discussion in the revised version.

Comments on revised version:Although the authors have responded to my concerns as they deemed suitable, these concerns still stand for the revised version.

Given our responses above and the lack of detail in this comment, we are unsure why the Reviewer is still concerned.

**Reviewer #3 (Public review):**
Summary:This work is a detailed and thorough analysis of the morphogenesis of the posterior signaling center (PSC), a hematopoietic niche in the *Drosophila* larva. Live imaging is performed from the stage of PSC determination until the appearance of a compact lymph gland and PSC in the stage 16 embryo. This analysis is combined with genetic studies that clarify the involvement of adjacent tissue, including the visceral mesoderm, alary muscle, and cardioblasts/dorsal vessel. Lastly, the Slit/Robo signaling system is clearly implicated in the normal formation of the PSC.Strengths:The data are clearly presented and well documented, and fully support the conclusions drawn from the different experiments.The authors have addressed all of my previous comments, in particular concerning the role of epidermal cell rearrangements during dorsal closure as a possible force acting on the movement of PSC cells. The authors have clarified their definition of "collective migration" as it applies to the movement of PSC. The revised paper will make an important contribution to our understanding of the mechanisms driving morphogenesis.

We are appreciative of the time spent by the Reviewer reading our responses and assessing the revision.

---------

The following is the authors’ response to the original reviews.

**Public Reviews:**

**Reviewer #1 (Public Review):**
Summary:The study by Nelson et al. is focused on the formation of the *Drosophila* Posterior Signaling Center (PSC) which ultimately acts as a niche to support hematopoietic stem cells of the lymph gland (LG). Using a combination of genetics and live imaging, the authors show that PSC cells migrate as a tight collective and associate with multiple tissues during a trajectory that positions them at the posterior of the LG.This is an important study that identifies Slit-Robo signaling as a regulator of PSC morphogenesis, and highlights the complex relationship of interacting cell types - PSC, visceral mesoderm (VM), and cardioblasts (CBs) - in the coordinated development of these three tissues during organ development. However, one point requiring clarification is the idea that PSC cells exhibit a collective cell migration; it is not clear that the cells are migrating rather than being pushed to a more dorsal position through dorsal closure and/or other similar large-scale embryo movement. This does not detract from the very interesting analysis of PSC morphogenesis as presented.

Since each referee asked for clarification concerning collective cell migration, we present a combined response further below, placed after the comments from Reviewer #3.

Strengths:(1) Using the expression of Hid or Grim to ablate associated tissues, they find evidence that the VM and CB of the dorsal vessel affect PSC migration/morphology whereas the alary muscles do not. Slit is expressed by both VM and CBs, and therefore Slit-Robo signaling was investigated as PSCs express Robo.(2) Using a combination of approaches, the authors convincingly demonstrate that Slit expression in the CBs and VM acts to support PSC positioning. A strength is the ability to knockdown slit levels in particular tissue types using the Gal4 system and RNAi.(3) Although in the analysis of robo mutants, the PSC positioning phenotype is weaker in the individual mutants (robo1 and robo2) with only the double mutant (robo1,robo2) exhibiting a phenotype comparable to the slit RNAi. The authors make a reasonable argument that Slit-Robo signaling has an intrinsic effect, likely acting within PSCs because PSCs show a phenotype even when CBs do not (Figure 4G).(4) New insight into dorsal vessel formation by VM is presented in Figure 4A, B, as loss of the VM can affect dorsal vessel morphogenesis. This result additionally points to the VM as important.Weaknesses:(1) The authors are cautioned to temper the result that Slit-Robo signaling is intrinsic to PSC since the loss of robo may affect other cell types (besides CBs and PSCs) to indirectly affect PSC migration/morphogenesis. In fact, in the robo2, robo1 mutant, the VM appears to be incorrectly positioned (Figure 4G).

We have reexamined our wording in the relevant Results section and, given that this referee agrees that we, “make a reasonable argument that Slit-Robo signaling has an intrinsic effect, likely acting within PSCs because PSCs show a phenotype even when CBs do not (Figure 4G)”, it was not clear how we might temper our conclusions more. Given that PSC cells express Robo1 and Robo2, and that the Vm does not contact the PSC, our ‘reasonable argument’ appears fair and parsimonious. Since we agree with the referee that a reader should be made as aware as possible of alternatives, we will add a comment to the Discussion, reminding the reader of the possibility of a secondary defect.

(2) If possible, the authors should use RNAi to knockdown Robo1 and Robo2 levels specifically in the PSCs if a Gal4 is available; might Antp.Gal4 (Fig 1K) be useful? Even if knockdown is achieved in PSCs+CBs, this would be a better/complementary experiment to support the approach outlined in Figure 4D.

While we agree that PSC-specific knockdown of Robo1 and Robo2 simultaneously would be ideal, this is not possible. First, the most-effective UAS-RNAi transgenes (that is, those in a Valium 20 backbone) are both integrated at the same chromosomal position; these cannot be simultaneously crossed with a GAL4 transgenic line to attempt double knock down. Additionally, as with all RNAi approaches that must rely on efficient knockdown over the rapid embryonic period, even having facile access to the above does not ensure the RNAi approach will cause as effective depletion as the genetic null condition that we use. Second, as the referee concedes, there is no embryonic PSC-specific GAL4. The proposed use of Antp-GAL4 would cause knockdown in many tissues (PSC, CB, Vm, epidermis and amnioserosa). This would lead to a reservation similar to that caused by our use of the straight genetic double mutant, as regards potential indirect requirement for Robo function.

(3) Movies are hard to interpret, as it seems unclear that the PSCs actively migrate rather than being pushed/moved indirectly due to association with VM and CBs/dorsal vessel.

First, the Vm does not directly contact the PSC, so it cannot be pushing the PSC dorsally. We will re-examine our text to be certain to make this clear. Second, in our analysis of *bin* mutants, which lack Vm, LGs and PSCs are able to reach the dorsal midline region in the absence of Vm. Finally, please see our response to Reviewer #3, point 2, for why we maintain that PSC cells are “migrating” even though some PSC cells are attached to CBs.

**Reviewer #2 (Public Review):**
The paper by Nelson KA, et al. explored the collective migration, coalescence, and positioning of the posterior signaling center (PSC) cells in *Drosophila* embryo. With live imaging, the authors observed the dynamic progress of PSC migration. Throughout this process, visceral mesoderm (VM), alary muscles (Ams), and cardioblasts (CBs) are in proximity to PSC. Genetic ablation of these tissues reveals the requirement for VM and CBs, but not AMs in this process. Genetic manipulations further demonstrated that Slit-Robo signaling was critical during PSC migration and positioning. While the genetic mechanisms of positioning the PSC were explored in much detail, including using live imaging, the functional consequence of mispositioning or (partial) absence of PSC cells has not been addressed, but would much increase the relevance of their findings. A few additional issues need to be addressed as well in this otherwise well-done study.Major points:(1) The only readout in their experiments is the relative correctness of PSC positioning. Importantly, what is the functional consequence if PSC is not properly positioned? This would be particularly important with robo-sli manipulations, where the PSC is present but some cells are misplaced. What is the consequence? Are the LGs affected, like the specification of their cell types, structure, and function? To address this for at least the robo-slit requirement in the PSC, it may be important to manipulate them directly in the PSC with a split Gal4 system, using Antp and Odd promoters.

We agree that the functional consequence of PSC mis-positioning is important and a relevant question to eventually address. However, virtually all markers and reagents used to assess the effect of the PSC on progenitor cells and their differentiated descendants are restricted to analyses carried out on the third larval instar - some three days after the experiments reported here. Most of the manipulated conditions in our work are no longer viable at this phase and, thus, addressing the functional consequences of a malformed PSC will require the field to develop new tools.

As we noted in the Introduction, the consistency with which the wildtype PSC forms as a coalesced collective at the posterior of the LG strongly suggests importance of its specific positioning and shape, as has now been found for other niches (citations in manuscript). Additionally, in the Discussion we mention the existence of a gap junction-dependent calcium signaling network in the PSC that is important for progenitor maintenance. Without continuity of this network amongst all PSC cells (under conditions of PSC mis-positioning), we strongly anticipate that the balance of progenitors to differentiated hemocytes will be mis-managed, either constitutively, and / or under immune challenge conditions.

Finally, to our knowledge, the tools do not exist to build a “split Gal4 system using Antp and Odd promoters”. The expression pattern observed using the genomic Antp-GAL4 line must be driven by endogenous enhancers–none of which have been defined by the field, and thus cannot be used in constructing second order drivers. Similarly, for *odd skipped*, in the embryo the extant Odd-GAL4 driver expresses only in the epidermis, with no expression in the embryonic LG. Thus, the cis regulatory element controlling Odd expression in the embryonic LG is unknown. In the future, the discovery of an embryonic PSC-specific driver will aid in addressing the specific functional consequences of PSC mis-positioning.

(2) The densely, parallel aligned fibers in the part of Figure 1J seemed to be visceral mesoderm, but further up (dorsally) that may be epidermis. It is possible that the PSC migrate together with the epidermis? This should be addressed.

See response to Reviewer #3.

(3) Although the authors described the standards of assessing PSC positioning as "normal" or "abnormal", it is rather subtle at times and variable in the mutant or KD/OE examples. The criteria should be more clearly delineated and analyzed double-blind, also since this is the only readout. Further examples of abnormal positioning in supplementary figures would also help.

We appreciate the Reviewer’s concern and acknowledge that the phenotypes we observed were indeed variable, and, at times subtle. As we show and discuss in the paper, our results revealed that the signaling requirements for proper PSC positioning are complex; this was favorably commented upon by Reviewer #1 (“...highlights the complex relationship of interacting cell types - PSC, visceral mesoderm (VM), and cardioblasts (CBs) - in the coordinated development of these three tissues during organ development.…”). We suspect the phenotypic variability is attributable to any number of biological differences such as heterogeneity of PSC cells and an accompanying difference in the timing of their competence to receive and respond to Slit-Robo signaling, the timing of release of Slit from CBs and Vm, number of cells in a given PSC, which PSC cells in the cluster respond to too little or too much signaling, and/or typical variability between organisms. Furthermore, PSC positioning analyses were conducted by two of the authors, who independently came to the same conclusions. For many of the manipulations double blinding was not possible since the genotype of the embryo was discernible due to the obvious phenotype of the manipulated tissue.

(4) The Discussion is very lengthy and should shortened.

We will re-examine the prose and emphasize more conciseness, while maintaining clarity for the reader.

**Reviewer #3 (Public Review):**
Summary:This work is a detailed and thorough analysis of the morphogenesis of the posterior signaling center (PSC), a hematopoietic niche in the *Drosophila* larva. Live imaging is performed from the stage of PSC determination until the appearance of a compact lymph gland and PSC in the stage 16 embryo. This analysis is combined with genetic studies that clarify the involvement of adjacent tissue, including the visceral mesoderm, alary muscle, and cardioblasts/dorsal vessels. Lastly, the Slit/Robo signaling system is clearly implicated in the normal formation of the PSC.Strengths:The data are clearly presented, well documented, and fully support the conclusions drawn from the different experiments. The manuscript differs in character from the mainstay of "big data" papers (for example, no sets of single-cell RNAseq data of, for instance, PSC cells with more or less Slit input, are offered), but what it lacks in this regard, it makes up in carefully planned and executed visualizations and genetic manipulations.Weaknesses:A few suggestions concerning improvement of the way the story is told and contextualized.(1) The minute cluster of PSC progenitors (5 or so cells per side) is embedded (as known before and shown nicely in this study) in other "migrating" cell pools, like the cardioblasts, pericardial cells, lymph gland progenitors, alary muscle progenitors. These all appear to move more or less synchronously. What should also be mentioned is another tissue, the dorsal epidermis, which also "moves" (better: stretches?) towards the dorsal midline during dorsal closure. Would it be reasonable to speculate (based on previously published data) that without the force of dorsal closure, operating in the epidermis, at least the lateral>medial component of the "migration" of the PSC (and neighboring tissues) would be missing? If dorsal closure is blocked, do essential components of PSC and lymph gland morphogenesis (except for the coming-together of the left and right halves) still occur? Are there any published data on this?

Each of the Reviewers is interested in our response to this very relevant question, and, thus, we will address the issue en bloc here. First, we will add a Supplementary Figure showing that LG and CBs are still able to progress medially towards the dorsal midline when dorsal closure stalls. This rules out any major effect for the most prominent “large-scale embryo cell sheet movement” in positioning the PSC. Second, published work by Haack et. al. and Balaghi et. al. shows that CBs and leading edge epidermal cells are independently migratory, and we will add this context to the manuscript for the reader.

(2) Along similar lines: the process of PSC formation is characterized as "migration". To be fair: the authors bring up the possibility that some of the phenotypes they observe could be "passive"/secondary: "Thus, it became important to test whether all PSC phenotypes might be 'passive', explained by PSC attachment to a malforming dorsal vessel. Alternatively, the PSC defects could reflect a requirement for Robo activation directly in PSC cells." And the issue is resolved satisfactorily. But more generally, "cell migration" implies active displacement (by cytoskeletal forces) of cells relative to a substrate or to their neighbors (like for example migration of hemocytes). This to me doesn't seem really clearly to happen here for the dorsal mesodermal structures. Couldn't one rather characterize the assembly of PSC, lymph gland, pericardial cells, and dorsal vessel in terms of differential adhesion, on top of a more general adhesion of cells to each other and the epidermis, and then dorsal closure as a driving force for cell displacement? The authors should bring in the published literature to provide a background that does (or does not) justify the term "migration".

Before addressing this specifically, we remind readers of our response above that states the rationale ruling out large, embryo-scale movements, such as epidermal dorsal closure, in driving PSC positioning. So, how are PSC cells arriving at their reproducible position? This manuscript reports the first live-imaging of the PSC as it comes to be positioned in the embryo. We interpret these movies to suggest strongly that these cells are a ‘collective’ that migrates. Neither the data, nor we, are asserting that each PSC cell is ‘individually’ migrating to its final position. Rather, our data suggest that the PSC migrates as a collective. The most paradigmatic example of directed, collective cell migration, is of *Drosophila* ovarian border cells. That cell cluster is surrounded at all times by other cells (nurse cells, in that case), and for the collective to traverse through the tissue, the process requires constant remodeling of associations amongst the migrating cells in the collective (the border cells), as well as between cells in the collective and those outside of it (the nurse cells). In fact, the nurse cells are considered the substrate upon which border cells migrate. Note also that in collective border cell migration cells within the collective can switch neighbors, suggesting dynamic changes to cell associations and adhesions.

In our analysis, the PSC cells exhibit qualities reminiscent of the border cells, and thus we infer that the PSC constitutes a migratory cell collective. We also show in Figure 1H that PSC cells exhibit cellular extensions, and thus have a very active, intrinsic actin-based cytoskeleton. In fact, in Figure 1I, we point out that PSC cells shift position within the collective, which is not only a direct feature of migration, but also occurs within the border cell collective as that collective migrates. Additionally, the fact that the lateral-most PSC cells shift position in the collective while remaining a part of the collective–and they do this while executing net directional movement–makes a strong argument that the PSC is migratory, as no cell types other than PSCs are contacting the surfaces of those shifting PSC cells. Lastly, the Reviewer’s supposition that, rather than migration, dorsal mesoderm structures form via “differential adhesion, on top of a more general adhesion of cells to each other” is, actually, precisely an inherent aspect of collective cell migration as summarized above for the ovarian border collective.

In our resubmission we will adjust text citing the existing literature to better put into context the reasoning for why PSC formation based on our data is an example of collective cell migration.

(3) That brings up the mechanistic centerpiece of this story, the Slit/Robo system. First: I suggest adding more detailed data from the study by Morin-Poulard et al 2016, in the Introduction, since these authors had already implicated Slit-Robo in PSC function and offered a concrete molecular mechanism: "vascular cells produce Slit that activates Robo receptors in the PSC. Robo activation controls proliferation and clustering of PSC cells by regulating Myc, and small GTPase and DE-cadherin activity, respectively". As stated in the Discussion: the mechanism of Slit/Robo action on the PSC in the embryo is likely different, since DE-cadherin is not expressed in the embryonic PSC; however, it maybe not be THAT different: it could also act on adhesion between PSC cells themselves and their neighbors. What are other adhesion proteins that appear in the late lateral mesodermal structures?

Could DN-cadherin or Fasciclins be involved?

We agree with the Reviewer that Slit-Robo signaling likely acts in part on the PSC by affecting PSC cell adhesion to each other and/or to CBs (lines 428-435). As stated in the Discussion, we do not observe Fasciclin III expression in the PSC until late stages when the PSC has already been positioned, suggesting that Fasciclin III is not an active player in PSC formation. Assessing whether the PSC expresses any other of the suite of potential cell adhesion molecules such as DN-Cadherin or other Fasciclins, and then study their potential involvement in the Slit-Robo pathway in PSC cells, would be part of a follow-up study.

**Recommendations for the authors:**

**Reviewing Editor Comments:**
The authors are encouraged to address several key issues and provide more explicit clarification when interpreting the behavior of the PSC cells as "migration." It is recommended that the authors engage with all reviewers' comments and refine the text based on the feedback they find valuable.
**Reviewer #1 (Recommendations For The Authors):**
Major concerns:(1) Is it possible to assay robo1 and/or robo1 RNAi in a tissue-specific manner to further explore an intrinsic role in the PSC? Might the VM indirectly affect PSCs in a CB-independent manner? How does this affect the interpretation of results in Figure 4.

See also our response to Reviewer #1, Public review weaknesses #2.

Though we agree with the Reviewer that this is the better experiment to test for an intrinsic role for Robo in the PSC, this experiment is not possible at this time. As we noted in the manuscript, we do not yet have an embryonic PSC-specific GAL4, though we have been putting efforts towards identifying/developing such a tool. The Antp-GAL4 driver we used in this study will drive not only in both PSCs and CBs, but also in Vm, epidermis, and amnioserosa, as well as other tissues. The other available embryonic PSC drivers are not specific to the PSC and will drive expression in CBs and Vm, at minimum. This, combined with the reality that RNAi can be ineffective in embryonic tissues, resulted in our use of whole organism mutants to best address this question.

We acknowledge that it is possible the Vm indirectly effects the PSC in a CB-independent manner in the double Robo mutant, and we added a statement to the Discussion reiterating this point. However, because the PSC expresses Robo1 and Robo2, we maintain that the simplest interpretation of the results in Figure 4 is that PSC cells require intrinsic Robo signaling. And, as we state in the manuscript, it is possible that Slit signals directly from Vm to Robo on the PSC.

(2) As this is the first study to be presenting PSC formation as involving collective cell migration, can the authors provide experimental evidence and rationale for this categorization?

We have added our rationale to the Results section in the revision.

See also our response to Reviewer #3, Public review weakness #2.

(3) The Slit staining presented in Fig 3 W', Z' should be quantified. Furthermore, what is the VM phenotype when Robo1 is overexpressed? Is there a VM-specific phenotype and could this indirect effect cause the PSC to misform/mismigrate?

We didn’t quantify Slit levels in the Vm-specific Robo overexpression condition because there was a visually striking difference compared to controls (increased intensity and specific localization to Vm membranes), and the manipulation resulted in a PSC phenotype. Thus, the evidence we show appears sufficient to strongly suggest that our genetic manipulation resulted in successful trapping of Slit on the Vm.

As to a Vm phenotype when Robo1 is overexpressed Vm-specifically: we know Vm is present, but we haven’t performed an in-depth phenotypic analysis. In the manuscript we show that this manipulation at least affects organization of PSC-adjacent CBs, which we go on to show is correlated with mis-positioned PSCs. Thus, the PSC phenotype in this condition is not solely due to a Vm-specific phenotype.

Minor concerns/suggestions:(1) I might have missed it but where are the Movies referenced in the text? Are legends provided for the videos? It is important that this is included in the final version (or more clearly presented if I missed it).

We thank you the Reviewer for pointing this out; we now direct the reader to the movies at appropriate places within the text.

(2) In Figure 5, it might be helpful to add a third column to A in which the PSCs are pseudo-colored and thus highlighted because it is difficult to discern the white (not pink) PSCs...

We appreciate the suggestion and now include these panels as Figure 5A’’ in the revision.

(3) If I am following correctly, the lost PSC cells in Figure 5 don't move. Doesn't this suggest that what is critical is that the PSCs attach to the VM and/or CBs, and not necessarily that they are an actively migrating cell type? They "move" but might be passively carried.

See also the response to Reviewer #3, Public reviews weaknesses #2.

The Reviewer is correct that the PSC cells in Fig. 5 don’t move very much, but we interpret this differently from the Reviewer. After detachment of the cells in question they undergo dramatic shape changes, indicating active cytoskeletal remodeling, so the molecular machinery needed for migration appears to remain intact. Thus, we suggest that this observation actually emphasizes our finding that collectivity is needed for the migration. Given the consistency of PSC coalescence/collectivity and the intricate regulation that controls it, we believe it to be an integral part of PSC identity. When PSC cells become detached, they likely lose an aspect of their identity. In various manipulations we’ve noted instances of severely dispersed PSC cells expressing very low levels of identity markers Antp or Odd. Cells in such cases are likely compromised for their function, and this can include, for example, whether they can properly sense cues for migration.

**Reviewer #2 (Recommendations For The Authors):**
Minor points:(1) The expression pattern of Antp-Gal4 > myrGFP in the whole embryo should be shown to better demonstrate the overlap with Odd. How does it compare with Antp-Gal4 > CD8::GFP?

We do not understand the question posed. We are not suggesting that Antp and Odd overlap in all cells, nor even many cells. It has been demonstrated by the field that co-expression among mesodermal cells, in the position where LG cells are specified, is a marker for the PSC. We have not thoroughly investigated all reporter lines for the GAL4 drivers used by the field.

(2) Does Tincdelta4-Gal4 not at all express in the PSC? This should be verified.

This question appears to refer to depletion of Slit by RNAi or cell killing driven by tinCΔ4-GAL4. TinCΔ4-GAL4 is expressed in CBs and in precisely 1 embryonic PSC cell. First, Slit isn’t expressed by any PSC cells to our eye, so any PSC mis-positioning observed upon tinCΔ4>Sli RNAi implicates CB involvement in PSC positioning. In designing tests for CB involvement, we were unable to identify any mutant known to lack CBs (or have fewer CBs) that didn’t also affect specification of the LG/PSC. The cell killing approach seemed best. It is possible that, in this scenario, perhaps ablation of a single, key PSC cell could affect final positioning of the other PSCs, but we think that less likely than a role for CBs. We also retain our original conclusion due to the fact that we often find mis-positioned PSC cells adjacent to mis-positioned CBs, including in the panel representing the CB ablation experiment, Figure 2S.

(3) Line 212: The data provide evidence that Vm is necessary, but clearly not sufficient, as CBs are also necessary.

We see how this wording was misleading and have adjusted the text accordingly.

(4) The CBs are not visible in Figure 3B.

We are unsure what the Reviewer is referring to, as we are certain that the signal between the blue outlines is indeed Slit expression in CBs.

**Reviewer #3 (Recommendations For The Authors):**
One minor mistake (I believe): in line 229 it should say "3C and 3D"

We have corrected this error.